# RNF19A-mediated ubiquitination of BARD1 prevents BRCA1/BARD1-dependent homologous recombination

Qian Zhu [1,2,6], Jinzhou Huang [2,6], Hongyang Huang [3], Huan Li[1], Peiqiang Yi[1], Jake A. Kloeber [2,4], Jian Yuan [5], Yuping Chen[5], Min Deng [2], Kuntian Luo[2], Ming Gao[2], Guijie Guo[2], Xinyi Tu[2], Ping Yin[2], Yong Zhang[2], Jun Su[1], Jiayi Chen [1✉] & Zhenkun Lou [2✉]

BRCA1-BARD1 heterodimers act in multiple steps during homologous recombination (HR) to ensure the prompt repair of DNA double strand breaks. Dysfunction of the BRCA1 pathway enhances the therapeutic efficiency of poly-(ADP-ribose) polymerase inhibitors (PARPi) in cancers, but the molecular mechanisms underlying this sensitization to PARPi are not fully understood. Here, we show that cancer cell sensitivity to PARPi is promoted by the ring between ring fingers (RBR) protein RNF19A. We demonstrate that RNF19A suppresses HR by ubiquitinating BARD1, which leads to dissociation of BRCA1-BARD1 complex and exposure of a nuclear export sequence in BARD1 that is otherwise masked by BRCA1, resulting in the export of BARD1 to the cytoplasm. We provide evidence that high RNF19A expression in breast cancer compromises HR and increases sensitivity to PARPi. We propose that RNF19A modulates the cancer cell response to PARPi by negatively regulating the BRCA1-BARD1 complex and inhibiting HR-mediated DNA repair.

[1] Department of Radiation Oncology, Ruijin Hospital, Shanghai Jiaotong University School of Medicine, Shanghai 200025, China. [2] Department of Oncology, Mayo Clinic, Rochester, MN 55905, USA. [3] Department of Pathology, Li Ka Shing Faculty of Medicine, The University of Hong Kong, Hong Kong 999077, China. [4] Mayo Clinic Medical Scientist Training Program, Mayo Clinic, Rochester, MN 55905, USA. [5] Research Center for Translational Medicine, East Hospital, Tongji University School of medicine, Shanghai 200120, China. [6]These authors contributed equally: Qian Zhu, Jinzhou Huang. ✉email: chenjiayi0188@aliyun.com; Lou.Zhenkun@mayo.edu

The dutiful maintenance of genomic continuity and stability in organisms during DNA repair is critical for preventing the transformation of normal diploid cells to an oncogenic state[1,2]. In human cells, nonhomologous end joining (NHEJ) and homologous recombination (HR) are the two major pathways of double-strand breaks (DSBs) repair[3]. Compared with NHEJ, HR is a more error-free repair process that primarily functions in S/G2 phase[4], involving a coordinated series of complex steps composed of DNA-end resection, RAD51 filament arrangement on the resulting single-stranded DNA (ssDNA) to pair homologous sequence, heteroduplex formation, and resolution. Deficiency of HR induces cells to alternative, more error-prone DNA repair pathways, thus yielding genomic instability and cancer predisposition[5].

BRCA1, performing in concert with its obligatory partner BARD1, has multiple roles in both the initial and later stages of the HR process, which is essential for the timely repair of DSBs[6]. As the name implies, the BRCA1-associated RING domain-1 (BARD1) structurally links to BRCA1 through their conserved RING finger domains at the N-terminus, especially residues 1–109 of BRCA1 and residues 26–119 of BARD1, thereby carrying out various functions including DNA repair, substrate ubiquitination, and mRNA process regulation[7,8]. Both genes have been identified as tumor suppressors[9,10]. Collective evidence suggests that depletion or mutation of either BRCA1 or BARD1 is not only responsible for the development of familial breast and ovarian cancer but also various sporadic cancers[10–12]. Taking advantage of HR deficiency caused by functional inactivation or dissociation of the BRCA1/BARD1 complex has been regarded as an effective strategy to improve the therapeutic efficiency of poly (ADP-ribose) polymerase (PARP) inhibitors (PARPi)[13,14].

Although progress has been made in identifying these key factors and their underlying mechanistic roles, the post-translational modifications and their influence on the precise regulatory mechanisms of these proteins remain unclear. The ubiquitin system is best known for its role in proteolysis;[15] however, proteasome-independent ubiquitin signaling predominates in the DNA damage response (DDR)[16,17]. Therefore, we hypothesized that the stale interaction of BRCA1 and BARD1 might be controlled by ubiquitylation. Similar to p53, BRCA1 and BARD1 are proteins that shuttle between nuclear and cytoplasmic compartments[18,19]. For example, BARD1 contains a nuclear export sequence (NES) that is masked by BRCA1 binding[20]. This has functional implications since BRCA1/BARD1 must localize to the nucleus to participate in HR and suppress tumorigenesis, whereas their dissociation may lead to nuclear export and dysfunctional HR repair[21]. Previous studies used knockdown of BRCA1 or peptide competition to disrupt the BRCA1-BARD1 interaction and unmask the NES of BARD1[21,22]. However, it was not understood how the NES of BARD1 is unmasked under physiological conditions and how this is regulated. Therefore, elucidating the underlying mechanisms that regulate the BRCA1/BARD1 complex, will be useful to more directly study cellular consequences of this interaction. Protein ubiquitination has emerged as an important factor in the control of DDR pathways. RNF19A has previously been implicated in NF-κB signaling, fertilization, and neuroinflammation[23,24]. Emerging evidence has shown the association between RNF19A and cancers. For example, RNF19A mRNA was amplified in the blood of prostate cancer patients and RNF19A is aberrantly expressed in cancer-related fibroblast[25,26]. However, the detailed function of RNF19A in cancer remains unclear.

Here we identify that RNF19A, a ring between ring fingers (RBR) family E3 ligase, interacts with and ubiquitinates BARD1, resulting in the nuclear export of BARD1, thus compromising HR and sensitizing cancer cells to PARPi. Furthermore, clinical analysis indicates that the influence of RNF19A expression on prognosis is markedly affected by BARD1 levels. Our results elucidate how RNF19A-mediated ubiquitylation of BARD1 induces dissociation of BRCA1, unmasks the NES region of BARD1, restrains HR activity, and makes cancer cells more susceptible to PARPi treatment.

## Results

**RNF19A regulates HR upon DNA damage.** The RBR E3 ligases family comprises a group of 14 multi-domain enzymes[27], among which few have been analyzed in detail. Our group previously confirmed that the RBR protein Parkin is involved in the regulation of mitosis, necroptosis, and tumorigenesis[28,29]. We were interested in identifying the mechanism and functional significance of other E3 ligases in this group. We found a potential role of RNF19A in the DDR. We first examined γ-H2AX focus formation, a pan DNA damage marker, in parental and RNF19A-deficient cells exposed to ionizing radiation (IR). As shown in Fig. 1a, b, depletion of RNF19A resulted in decreased accumulation of γH2AX foci at late time points (8 h and 24 h). In contrast, overexpression of RNF19A induced sustained γH2AX foci even 8 h and 24 h after IR compared with control cells (Supplementary Fig. 1a, b). Also, decreased chromosomal breaks were observed in metaphase spreads from IR-treated, RNF19A knockout (KO) compared with control cells (Supplementary Fig. 1c, d). These results suggest RNF19A impairs DNA damage repair.

To further test whether RNF19A has a role in the DDR, we knocked down RNF19A and found cancer cells were more resistant to DNA-damaging agents, including Olaparib, Cisplatin, and IR (Fig. 1c–e), whereas ectopically expressed WT RNF19A reversed this phenomenon (Supplementary Fig. 1e–g). We next employed a well-established dual reporter assay for the simultaneous measurement of both HR and NHEJ[30] to examine whether and how RNF19A regulates DSBs repair. As shown in Fig. 1f, g, RNF19A deficiency increased HR efficiency while NHEJ was mildly compromised. The I-SceI-based assay also exhibited an enhanced HR efficiency in RNF19A-depleted cells (Supplementary Fig. 1h). Importantly, we did not find significant changes in the cell-cycle profile in either RNF19A deficiency cells or in cells rescued with a WT RNF19A construct (Supplementary Fig. 1i), suggesting the HR efficiency alteration by RNF19A was not due to an indirect effect of cell-cycle change). These data suggest that RNF19A inhibits HR repair.

To identify potential targets of RNF19A in HR, we first examined the ability of the main DDR factors to form foci following damage. The response to DSBs starts with the kinase ATM phosphorylating MDC1, which then recognizes phosphorylated histone H2AX (γH2AX) and amplifies the damage response. The ubiquitin (Ub) signaling is then activated and recruits repair proteins such as BRCA1 and 53BP1 to chromatin surrounding DSBs, which are involved in HR and NHEJ, respectively[31]. RNF19A did not influence the focus formation of upstream regulators involved in DDR, such as γ-H2AX (1 h, Fig. 1a and Supplementary Fig. 1a), MDC1, and FK2 (Ub) (Fig. 1h, i). RNF19A also had no effect on 53BP1 recruitment to DSBs (Fig. 1h, i), suggesting it is not related to the NHEJ pathway. On the other hand, downregulation of RNF19A resulted in a significantly elevated accumulation of BRCA1, BARD1, RAD51, and RPA32 focus formation (Fig. 1h, i). Furthermore, compromised accumulation of BRCA1/BARD1 foci was observed in RNF19A-overexpressed cells at both early and late time points (Supplementary Fig. 1j–l). Taken together, these results suggest that RNF19A regulates HR repair by inhibiting BRCA1/BARD1 recruitment to DSB sites.

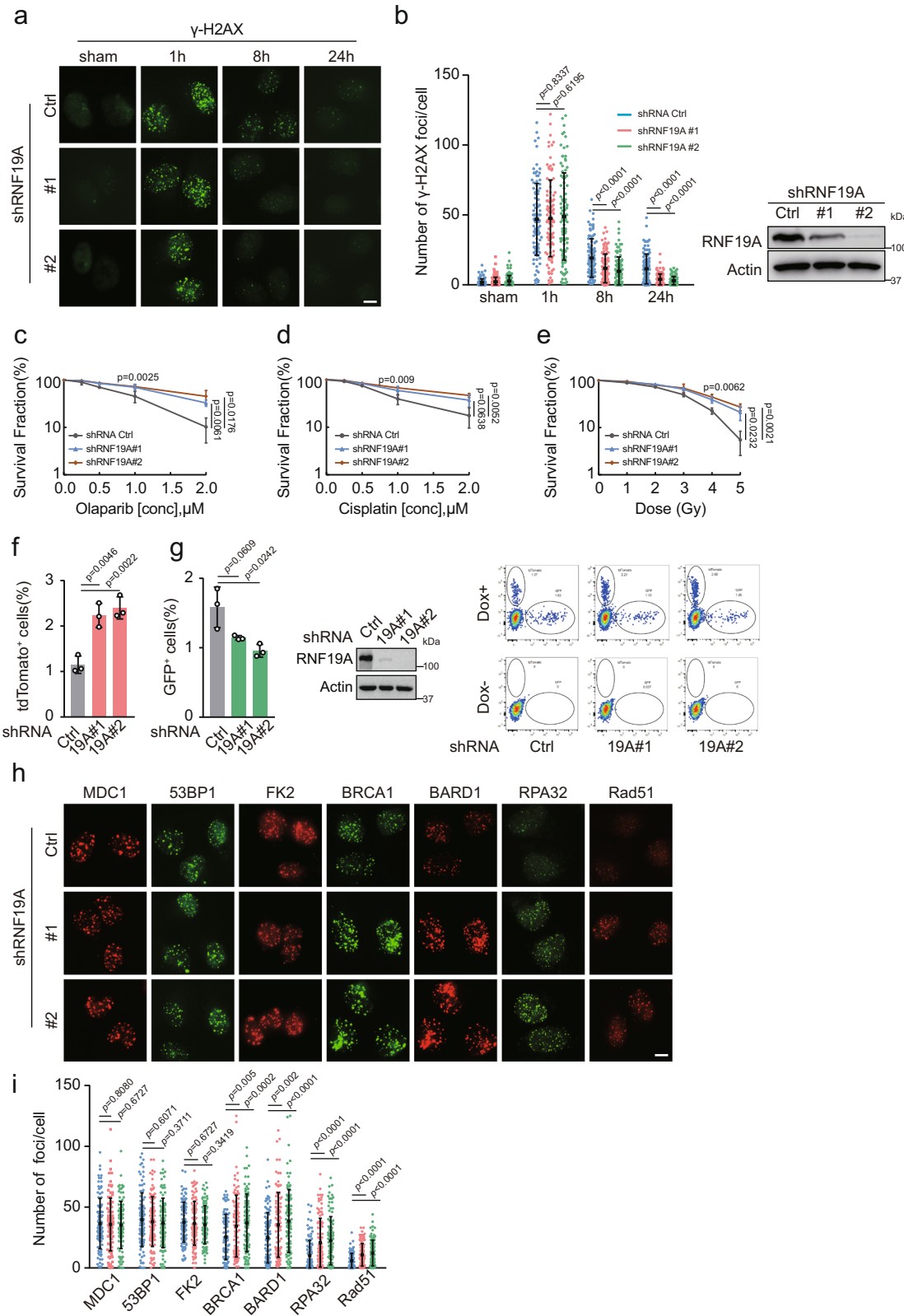

**RNF19A interacts with BARD1 via its RING1 domain**. As RNF19A affects BRCA1/BARD1 focus formation, we further examined whether there is a crosstalk between RNF19A and the BRCA1/BARD1 complex. Interestingly, we found that RNF19A interacted with BARD1 but not BRCA1 (Fig. 2a–c and Supplementary Fig. 2a). We next mapped which region of RNF19A is responsible for BARD1 interaction. The RBR domains form specifically ordered interactions with RING1 sequentially followed by IBR (In-between RING) and RING2. The RING1 domain binds E2 and ubiquitin is transferred to a specific Cys residue within the RING2 domain[27,32]. The results in Fig. 2d, e showed that the RING1 domain of RNF19A is required for its interaction with BARD1. We also generated several deletion mutants of BARD1 to map the domain that interacts with

**Fig. 1 RNF19A inhibits HR and increases sensitivity to DNA-damaging agents. a–b** Control and RNF19A knockdown U2OS cells were treated with or without IR (2 Gy), γ-H2AX foci before or 1 h, 8 h, and 24 h after IR was detected by immunofluorescence. Nuclei were visualized with DAPI (blue). Representative images are shown in **a**. Quantification of focus signals per cell (each dot represents a single cell, $n = 100$) is shown in **b**. Error bars represent means ± s.d. of three independent experiments. Scale bars, 10 μM. **c–e** The sensitivity of control (Ctrl) and RNF19A knockdown U2OS cells to Olaparib **c** cisplatin **d** and IR **e** was assessed by colony formation assay. Error bars are means ± s.d. of three independent experiments. **f–g** Control and RNF19A knockdown Clz3 cells, which contains a dual reporter for HR-tdTomato and NHEJ-GFP, were treated with doxycycline (Dox) for 48 h to turn on I-SceI expression and to induce DSBs. Cells were harvested and subjected to FACS analysis. Error bars are means ± s.d. of three independent experiments. **h–i** Control (Ctrl) or RNF19A knockdown U2OS cells were treated with IR (1 Gy, 1 h for MDC1, 53BP1, FK2, BRCA1, BARD1; 1 Gy, 5 h for RAD51 and 1 Gy, 3 h for RPA32), and indicated foci were detected by immunofluorescence. Nuclei were visualized with DAPI (blue). Representative images are shown in **h**. Quantification of focus signals per cell (each dot represents a single cell, $n = 100$) is shown in **i**. Error bars represent means ± s.d. of three independent experiments. Scale bars, 10 μM. $p$ values are determined by unpaired two-sided $t$ test in **b–g** and **i**. Source data are provided as a Source Data file.

RNF19A. The RING domain of BARD1 at the N-terminus mediates its dimerization with BRCA1 and the BRCA1 carboxy-terminal (BRCT) domain at BARD1's C-terminus can interact with various proteins such as HP1 and PAR. In addition, BARD1 has three ankyrin (ANK) repeats located upstream of the BRCT domain. This combination of RING, ANK, and BRCT domains is a unique feature of BARD1[10]. As shown in Fig. 2f, g, the RING domain of BARD1 is required for the BARD1/RNF19A interaction. To further investigate whether RNF19A interacts with BARD1, in vitro GST pull-down assays were performed using GST-RNF19A and His-BARD1 proteins purified from bacteria. The results showed that GST-RNF19A interacted with His-BARD, which was disrupted by the deletion of the RING1 domain of RNF19A (Fig. 2h). Taken together, these results indicate that RNF19A interacts with BARD1.

Importantly, RNF19A WT, but not RNF19A R1, was able to reverse the increase in HR repair caused by RNF19A deficiency (Fig. 2i) and re-sensitize cells to PARPi (Fig. 2j), suggesting that the RNF19A/BARD1 interaction is essential for HR regulation and cancer cell response to PARPi.

**RNF19A regulates DNA-end resection and HR through a BARD1-dependent manner.** Considering that RNF19A is an E3 ligase[23,33], we asked whether the regulation of HR by RNF19A is dependent on its catalytic activity. We reconstituted RNF19A-deficient cells with RNF19A-WT or a catalytically inactive mutant (CA). RNF19A-WT, but not RNF19A-CA or R3 (catalytic deletion domain of RNF19A in Fig. 2d), reversed the increased HR efficiency (Supplementary Fig. 3a, b) as well as cells' resistance to PARPi and IR that were caused by RNF19A deficiency (Supplementary Fig. 3c, d). The same trend was observed for increased BRCA1 and BARD1 foci (Supplementary Fig. 3e–g), whereas the initial induction of γ-H2AX was equal in cells overexpressing RNF19A-WT or CA (Supplementary Fig. 3h, i), suggesting the catalytic activity of RNF19A is critical for its regulation of HR. In addition, we found RNF19A-R2 (IBR domain deletion of RNF19A) could partially reverse the increase in HR repair caused by RNF19A deficiency (Supplementary Fig. 3j). Previous studies showed the structure of IBR domain is similar to RING2 and it might be involved in the transportation of E2~Ub[34,35]. As the IBR domain is not necessary for the BARD1/RNF19A interaction (Fig. 2e), we assumed that although RNF19A's IBR domain does not contain an active site Cys, it has a role in facilitating the E3 ligase activity of RING2. Therefore, deletion will restrict the enzymatic activity of RNF19A and have a mild impact on HR.

DNA-end resection and RAD51 nucleoprotein filament formation are two defining steps of HR. Both BRCA1 and BARD1 interact with RAD51 and their abrogation in mouse and human cells impairs RAD51 focus formation and HR efficiency[36]. The ssDNA tail generated by end resection will be occupied by the ssDNA-binding factor RPA32, which will further be replaced by RAD51 to initiate downstream repair events. Such exchange

on DNA is mediated by BRCA2-DSS1, whose recruitment to DNA damage sites and HR-mediated activity will be enhanced by BRCA1/BARD1[37], indicating BRCA1/BARD1 is involved in mediating the exchange of RPA32 with RAD51 on ssDNA.

Our results so far suggest that RNF19A affects the recruitment of BRCA1/BARD1 as well as RPA32 and RAD51 (Fig. 1h, i). We further employed RNF19A-deficient cells reconstituted with RNF19A WT or CA to examine these foci and found that RNF19A-WT, but not RNF19A-CA, could repress RPA32 and RAD51 focus formation (Fig. 3a–d). These results are in line with our hypothesis that RNF19A suppresses HR repair. Accordingly, we hypothesized that RNF19A regulates DNA-end resection through its effect on BRCA1/BARD1. We used the ER-AsiSI system to assess the resection efficiency, in which the restriction enzyme AsiSI is recruited to the nucleus induced by induction with 4-hydroxytamoxifen (4-OHT) treatment and generates DSBs at sequence-specific sites (Fig. 3e). As shown in Fig. 3f, depletion of RNF19A resulted in an apparent increase in the amount of ssDNA at all distances from DSBs, representing an enhanced resection capacity, and only RNF19A-WT reversed the increased efficiency of resection, further supporting the role of RNF19A in regulating HR through affecting end resection through its E3 ligase activity.

There is evidence that BRCA1 promotes DNA-end resection by acting as an antagonist of 53BP1 and regulating the MRE11-RAD50-NBS1 (MRN) complex and CtIP[38,39]. As an obligatory partner of BRCA1, BARD1 promotes retention of the BRCA1/BARD1 complex at DNA damage sites[40,41]. This retention is required for processes involved in HR, including RAD51 recruitment and DNA-end resection. Based on this, we additionally investigated how BARD1 affects end resection. Depletion of 53BP1 did not suppress the end resection defect in BARD1-deficient cells as BRCA1 deficiency (Supplementary Fig. 3k), which was consistent with reported studies that 53BP1 could partially overcome the HR defect in BRCA1-deficient but not in BARD1-deficient cells[36]. We also examined CtIP protein level and focus formation in BARD1 knockdown cells, and found that CtIP foci were extensively suppressed in BARD1-depleted cells compared to the control group (Supplementary Fig. 3l, m).

On the other hand, since end resection is a common step for HDR and single-strand annealing (SSA), whose defect will compromise both pathways[42], we examined SSA efficiency and found it was mildly increased in RNF19A-depleted cells (Supplementary Fig. 3o). An increased frequency of SCE in RNF19A-depleted cells also implied the enhancing HR activity compared with control cells (Supplementary Fig. 3p, q). These results indicated that RNF19A was involved in the regulation of end resection promoted by BARD1/BRCA1 at least partially through CtIP.

On the other hand, overexpression of RNF19A did not further affect HR (Fig. 3g), cells' sensitivity to PARPi (Fig. 3h), or radiotherapy (Supplementary Fig. 3r), as well as end resection

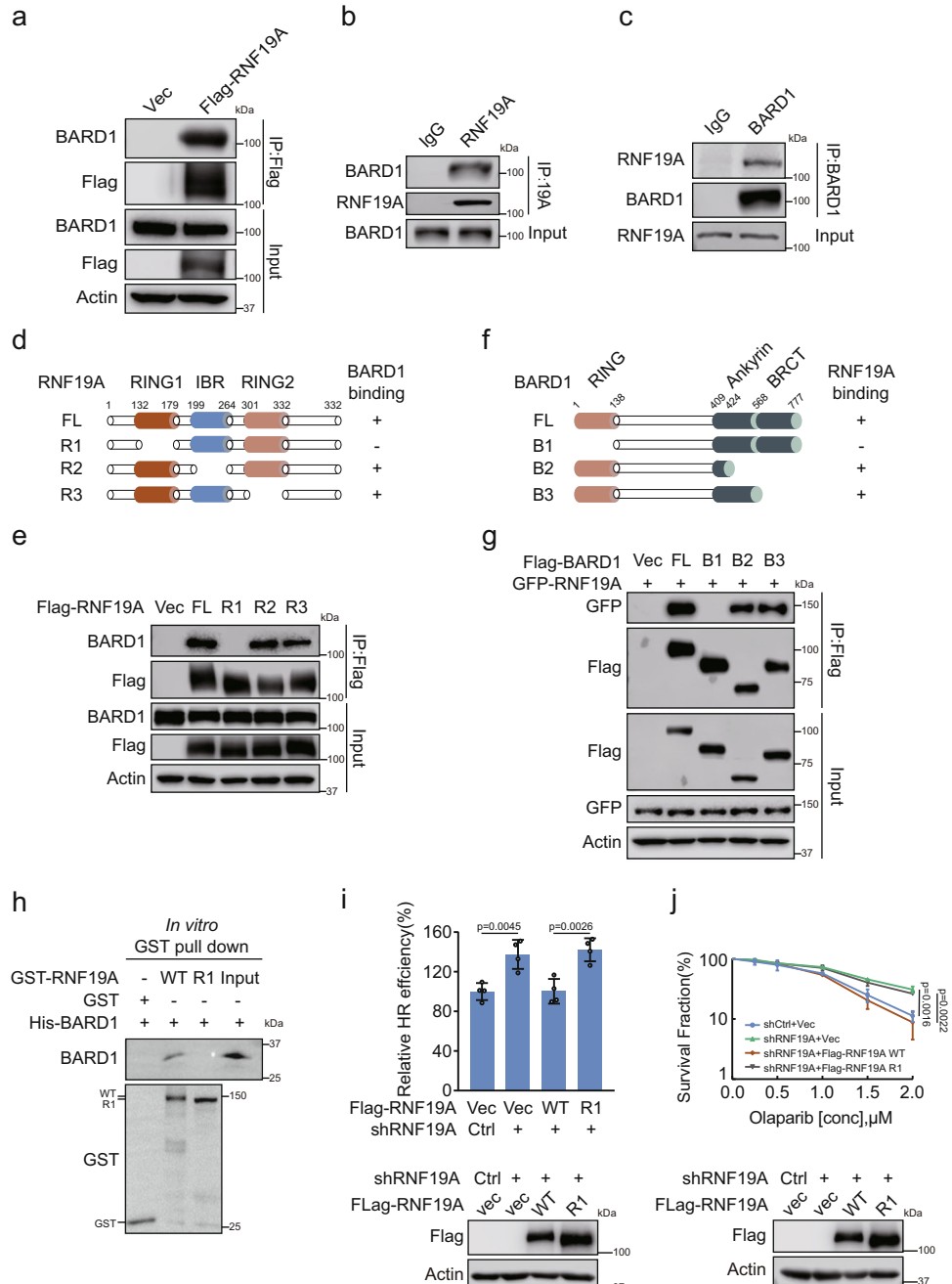

**Fig. 2 RNF19A interacts with the BARD1 RING domain through its RING2 region. a** HEK293T cells were transfected with Vec or Flag-RNF19A. Cell lysates were subjected to immunoprecipitation with Flag beads and immunoblotted with the indicated antibodies. **b–c** HEK293T cell lysates were subjected to immunoprecipitation with control IgG, RNF19A (**b**), or BARD1 (**c**) antibodies and immunoblotted with the indicated antibodies. **d–e** Diagram of RNF19A-WT and mutation constructs (**d**). R1, deletion of the RING1 domain (aa132–179); R2, deletion of the IBR domain (aa199–264); and R3, deletion of the RING2 domain (aa301–332). HEK293T cells transfected with Vec, WT, or deletion mutants of Flag-RNF19A outlined in (**d**) were subjected to immunoprecipitation as in **a**. **f–g** Diagram of BARD1-WT and mutation constructs (**f**). B1, deletion of the RING domain (aa1–138); B2, deletion of BRCT and parts of Ankyrin domain (aa424–777); and B3, deletion of the BRCT domain (aa568–777). HEK293T cells transfected with Vec, WT, or deletion mutants of Flag-BARD1 outlined in **f** together with GFP-RN19A were subjected to immunoprecipitation with Flag beads and immunoblotted with the indicated antibodies.
**h** Interaction between RNF19A and BARD1 was analyzed by in vitro GST pull-down assays using purified GST-RNF19A and His-BARD1 proteins. **i** Control (Ctrl) and RNF19A knockdown HEK293T cells stably expressing Vec, WT, or R1 Flag-RNF19A were subjected to DR-GFP-based HR assay. **j** Control (Ctrl) and RNF19A knockdown U2OS cells stably expressing Vec, WT, or R1 Flag-RNF19A were subjected to colony formation assay for assessment of the sensitivity to Olaparib. Error bars represent means ± s.d. of four (**i**) or three (**j**) independent experiments. *p* values are determined by unpaired two-sided *t* test in **i** and **j**. Source data are provided as a Source Data file.

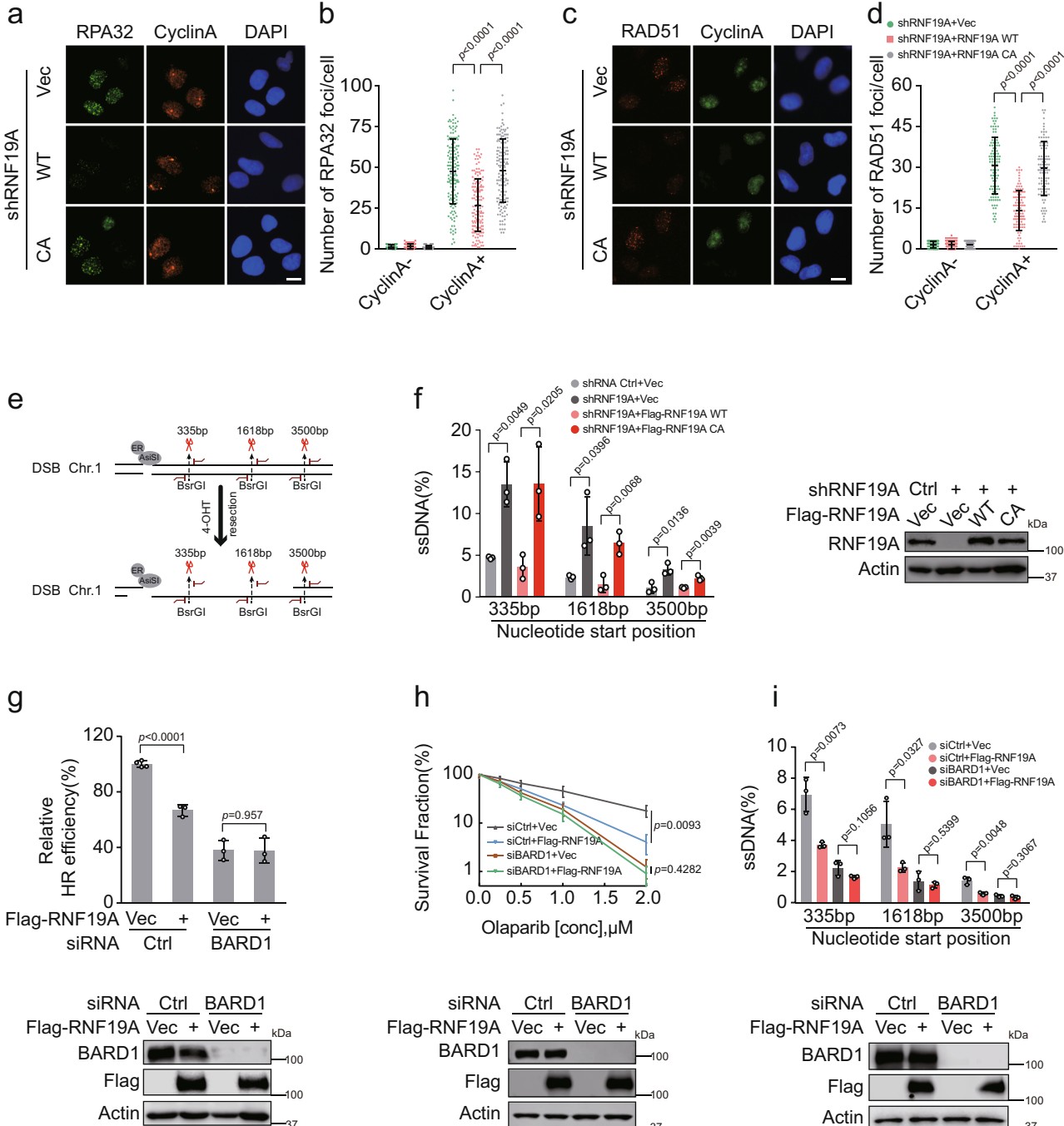

**Fig. 3 RNF19A functions in end resection through its catalytic activity and in a BARD1-dependent manner. a–d** RNF19A knockdown U2OS cells stably expressing Vec, WT, or C316A(CA) Flag-RNF19A were treated with IR (2 Gy, 3 h for RPA32 and 2 Gy, 5 h for RAD51). Cells were fixed and immunostained as indicated (**a** and **c**). Cyclin A was used as a marker of S/G2 phase. Quantification of focus signals per cell (each dot represents a single cell, $n = 100$ for group Cyclin A-; $n = 150$ in **b**; and $n = 110$ in **d** for group Cyclin A+) is shown in **b** and **d**. Scale bars, 10 μM. Error bars represent means ± s.d. of three independent experiments. **e** Schematic of ER-*Asi*SI system for quantification of DNA resection. Restriction enzyme *Asi*SI is fused to the estrogen receptor (ER) and can be induced to the nucleus and generate DSBs at sequence-specific sites by 4-OHT. The genomic DNA was extracted and quantification of ssDNA generated by resection was measured by qPCR. The primer pairs for DSBs are across BsrGI restriction sites. **f** RNF19A knockdown ER-*Asi*SI U2OS cells were reconstituted with Vec, WT, or C316A(CA) Flag-RNF19A and were pretreated with 300 nM 4-OHT for 4 h to induce DSBs. Genomic DNA was extracted and digested or mock digested with BsrGI overnight. DNA-end resection adjacent to indicated sites was measured by qPCR. **g** HEK293T cells stably expressing Vec or Flag-RNF19A were transfected with control (Ctrl) or BARD1 siRNAs for 48 h. HR efficiency was determined using the HR reporter assay. **h** U2OS cells stably expressing Vec or Flag-RNF19A were transfected with control (Ctrl) or BARD1 siRNAs were subjected to colony formation assay for assessment of response to Olaparib. **i** U2OS ER-*Asi*SI cells stably expressing Vec or Flag-RNF19A transfected with control (Ctrl) or BARD1 siRNAs for 48 h and were pretreated with 300 nM 4-OHT for 4 h before digest and measurement of DNA resection. Error bars represent means ± s.d. of three (**f**, **h**, and **i**) or four (**g**) independent experiments. *p* values are determined by unpaired two-sided *t* test in **b**, **d** and **f–i**. Source data are provided as a Source Data file.

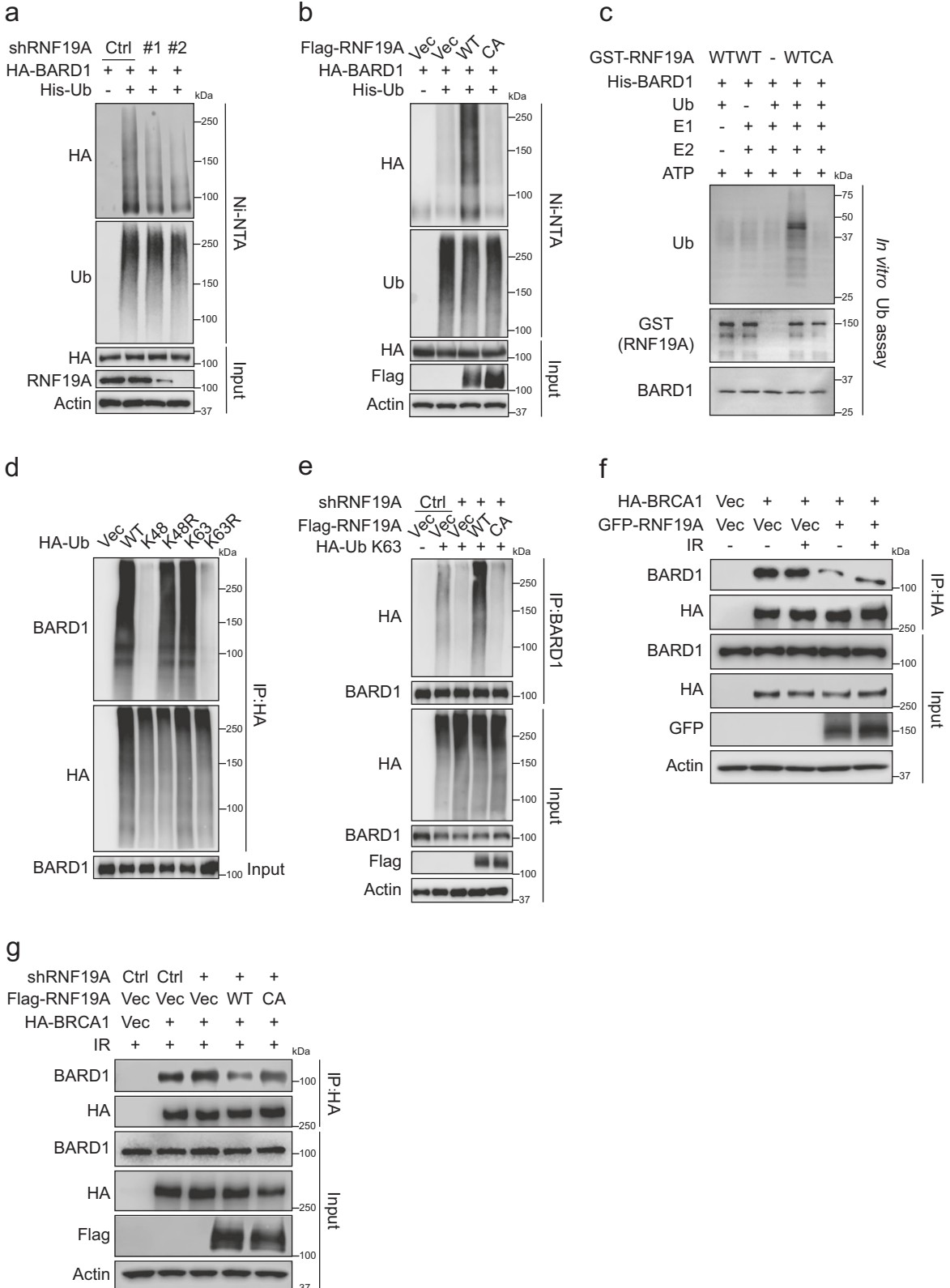

(Fig. 3i) in BARD1-deficient cells. Altogether, our results suggest that RNF19A regulates HR-mediated repair in a BARD1-dependent manner.

**RNF19A promotes ubiquitination of BARD1 and dissociation of the BRCA1/BARD1 interaction**. As RNF19A regulates resection and HR in a catalytic- and BARD1-dependent manner,

we questioned whether it can promote BARD1 ubiquitination. As shown in Fig. 4a, RNF19A deficiency diminished the ubiquitination of BARD1 in cells, while cells overexpressing RNF19A-WT, and not the CA mutant, elevated ubiquitination of BARD1 (Fig. 4b and Supplementary Fig. 4a). Similarly, overexpression of the catalytic deletion domain (R3 truncation in Fig. 2d) of RNF19A had no effect on BARD1 ubiquitination (Supplementary

**Fig. 4 RNF19A ubiquitinates BARD1 and restrains its interaction with BRCA1. a** Control or RNF19A knockdown HEK293T cells were transfected with indicated plasmids. Cell lysates were subjected to immunoprecipitation with His beads and immunoblotted with the indicated antibodies. **b** HEK293T cells were transfected with Vec, WT, or C316A(CA) Flag-RNF19A together with HA-BARD1 and His-Ub. Cell lysates were subjected to immunoprecipitation with His beads and immunoblotted with the indicated antibodies. **c** In vitro ubiquitination assay was performed by incubating purified GST-RNF19A and His-BARD1 (aa 26–327) proteins in the presence of recombinant E1, UbcH7, ubiquitin (Ub), and ATP buffer at 37 °C for 1 h. Samples were boiled and immunoblotted with indicated antibodies. **d** Indicated HA-tagged ubiquitin was transfected into HEK293T cells. Cell lysates were boiled, immunoprecipitated with HA beads, and immunoblotted with the indicated antibodies. **e** Control or RNF19A knockdown HEK293T cells were transfected with indicated plasmids. Cell lysates were boiled and immunoprecipitated with BARD1 antibody and immunoblotted with indicated antibodies. **f** HEK293T cells were transfected with indicated constructs and treated with or without IR (10 Gy, 1 h). Cell lysates were immunoprecipitated with HA beads and immunoblotted with indicated antibodies. **g** Control or RNF19A knockdown HEK293T cells were transfected with indicated constructs. Cell lysates were immunoprecipitated with HA beads and immunoblotted with indicated antibodies. Source data are provided as a Source Data file.

Fig. 4b), even though R3 is able to bind BARD1 (Fig. 2e). The ubiquitination of BARD1 by RNF19A was further confirmed by in vitro ubiquitination assays using purified GST-RNF19A and His-BARD1 proteins. WT GST-RNF19A promoted His-BARD1 ubiquitination in the presence of recombinant E1, E2 (UbcH7), and ubiquitin in vitro. The C316A mutation markedly reduced His-BARD1 ubiquitination in vitro (Fig. 4c). We also found that the K63-linked Ub chain was mainly responsible for BARD1 ubiquitination (Fig. 4d) and was greatly regulated by RNF19A (Fig. 4e). Meanwhile, RNF19A did not significantly influence BARD1 protein (Fig. 2a, e), as well as mRNA level (Supplementary Fig. 4c). These results suggest that RNF19A regulates BARD1 ubiquitination via the K63-linked Ub chain.

Most BARD1 and BRCA1 in cells form a functional heterodimer through their RING finger domains. This interaction is thought to stabilize both proteins and be important for DNA repair[18,43]. Expression of truncated BARD1 peptides incapable of interacting with BRCA1 results in HR dysfunction in both humans and mouse[44]. Because we observed that RNF19A regulates BARD1 ubiquitination and BRCA1/BARD1 retention at DSBs, we next explored whether RNF19A would affect BRCA1/BARD1 interaction. As shown in Fig. 4f, g and Supplementary Fig. 4d, overexpression of RNF19A depressed BRCA1/BARD1 interaction in cells, and this effect are dependent on the E3 ligase activity of RNF19A. Previous studies have shown that BARD1 colocalizes with BRCA1 in S-phase of the cell-cycle[36,45]. Accordingly, we then assessed BRCA1/BARD1 interaction in G1 and S phase. We confirmed an enriched interaction between BRCA1 and BARD1 in S phase (little interaction between BRCA1 and BARD1 was detected in G1 phase), and found that overexpression of RNF19A markedly reduced their association in S phase (Supplementary Fig. 4e). Meanwhile, we did not find significant differences in the expression levels of RNF19A and BARD1 ubiquitination by RNF19A across the cell cycle (Supplementary Fig. 4f, g). Taken together, our results suggest that RNF19A promotes BARD1 ubiquitination and affects the interaction between BARD1 and BRCA1.

**BARD1 ubiquitination by RNF19A is important for the BRCA1/BARD1 interaction and DDR.** So far, we have shown that RNF19A is involved in BRCA1/BARD1 recruitment to DSBs and performing HR. In addition to observing declined aggregation of BRCA1/BARD1 foci and decreased BRCA1/BARD1 interaction, we detected a stronger cytoplasmic fraction of BARD1, but not BRCA1 (although nuclear BRCA1 was decreased, resulting in a small decrease in nuc/cyto ratio), upon RNF19A overexpression (Supplementary Fig. 5a–c). We further confirmed this result by permeabilizing cells with saponin (0.02%) instead of regularly used triton, which enables cells to present a cytoplasmic signal predominantly[46]. As shown in Supplementary Fig. 5d, e, RNF19A-overexpressed cells exhibited a stronger cytoplasmic BARD1 signal, while cytoplasmic

expression of BRCA1 did not change much (Supplementary Fig. 5d, f). Although RNF19A did not alter the total protein level of BARD1, it promoted translocation of the BARD1 pool to cytoplasm upon DNA damage (Supplementary Fig. 5g, h). These results suggest that RNF19A is involved in nuclear-to-cytoplasmic shuttling of BARD1 in response to DNA damage. Previous studies have demonstrated that BRCA1 protein levels increase in late G1 and reach a maximum in S phase, while BARD1 is expressed at parallel levels throughout the cell cycle[47]. The dissociation of the heterodimer will lead to subcellular location or degradation of both proteins[20]. In contrast to BARD1, RNF19A-WT, but not RNF19A-CA, also led to a decrease of nuclear BRCA1 with no apparent increase of cytoplasmic BRCA1 (Supplementary Fig. 5i). These results suggest that the dissociation of BRCA1/BARD1 induced by RNF19A overexpression results in the nuclear export of BARD1, as well as BRCA1 instability.

We next mapped potential ubiquitination sites of BARD1 that were targeted by RNF19A and explored how BARD1 ubiquitination affects its binding to BRCA1. Similar to BRCA1, BARD1 has an NES within its RING domain. The NES region of BARD1 forms part of the BRCA1 dimerization domain, and the BRCA1/BARD1 interaction results in masking of the NES region and nuclear retention of BARD1[20]. Since we found that RNF19A can affect the BRCA1/BARD1 interaction and alter the location of BARD1, we suspected that the NES region is potentially targeted by RNF19A. We constructed a BARD1 truncation mutant with the NES deleted (aa 92–120) and found a greatly decreased RNF19A-mediated ubiquitination signal compared with WT (Fig. 5a), suggesting that the major ubiquitination sites might localize at this region. We then analyzed the BARD1 NES region and found two lysine residues-K96 and K110 that are conserved across several species[20]. We generated single and double mutations of these lysines (mutant K to R) and found that compared to BARD1-WT, single-site mutants (K96R and K110R) had a partial effect on the total ubiquitination, while the double K-to-R mutant almost abolished the ability of RNF19A to ubiquitinate BARD1(Fig. 5b), suggesting that both K96 and K110 of BARD1 are the major ubiquitin sites for RNF19A.

Next, we investigated the functional significance of BARD1 ubiquitination using BARD1-WT and the 2KR mutant. As shown in Fig. 5c, compared with BARD1-WT, BARD1-2KR had an enhanced association with BRCA1. Moreover, the association of BARD1-2KR and BRCA1 was not affected upon RNF19A overexpression (Fig. 5d). Collectively, these results suggest that BARD1 ubiquitination is pivotal for the regulation of BRCA1/ BARD interaction. We next examined whether BARD1 ubiquitination regulates its focus formation. Knocking down BARD1 using siRNAs targeting BARD1 almost abolished the focus formation of BARD1 and BRCA1 (Supplementary Fig. 5j). We then reconstituted cells with BARD1-WT or 2KR, and found both of them form foci normally in response to DNA damage.

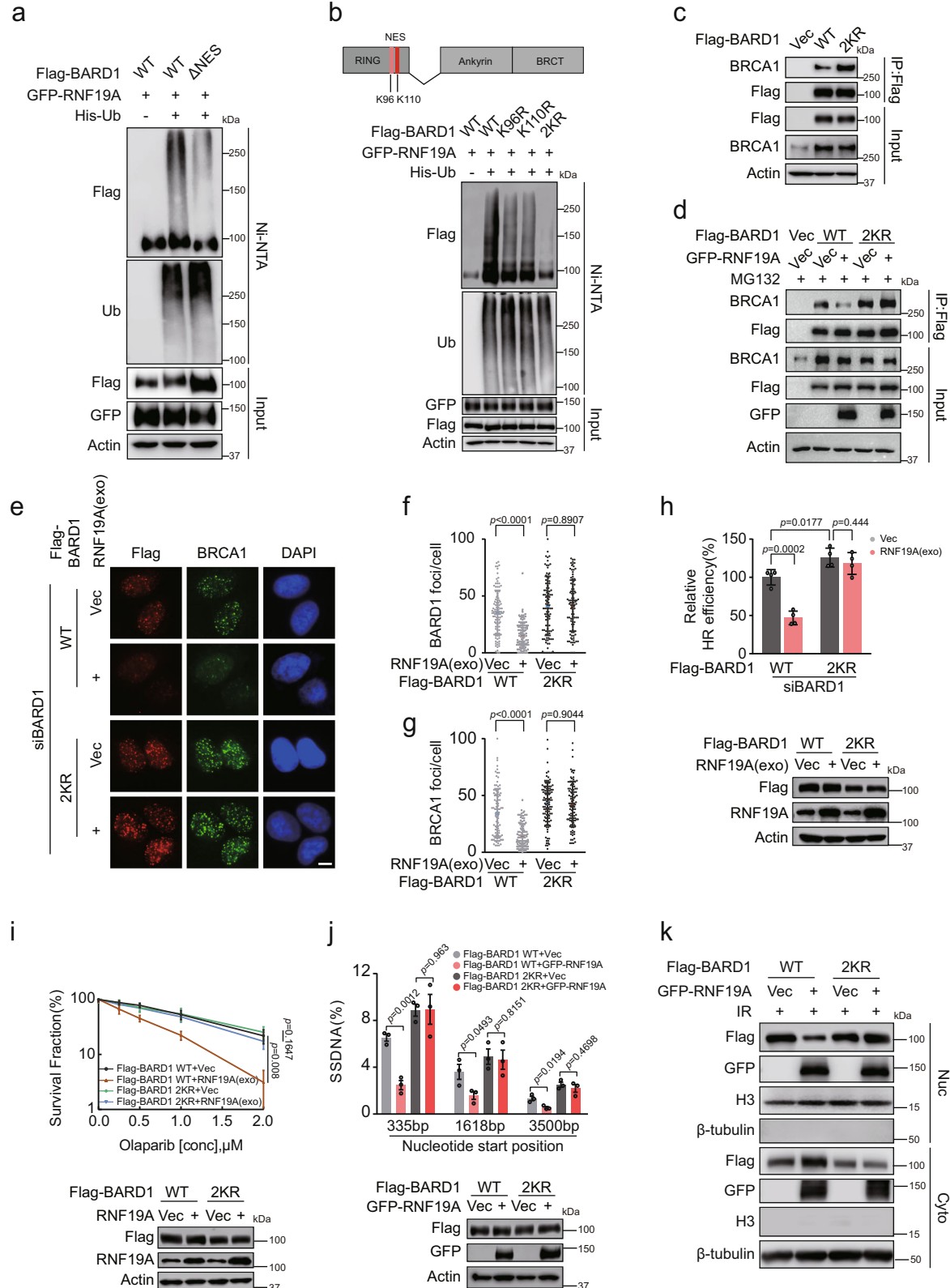

However, the focus accumulation of BARD1-WT was compromised in RNF19A-overexpressed cells, while the focus formation of BARD1-2KR was resistant to RNF19A regulation (Fig. 5e–g). BARD1-2KR did not form foci in the absence of DNA damage as BARD1-WT (Supplementary Fig. 5k), suggesting no premature DDR in the absence of BARD1 ubiquitination. Finally, we found that only BARD1-2KR was able to rescue RNF19A

overexpression-induced HR deficiency (Fig. 5h), cells' sensitivity to DNA damage agents (Fig. 5i and Supplementary Fig. 5l), as well as defects of DNA-end resection (Fig. 5j) and focus formation of RPA32 and RAD51 (Supplementary Fig. 5m–o). In addition, the nuclear-to-cytoplasmic translocation of BARD1-WT, but not BARD1-2KR, was largely influenced by RNF19A (Fig. 5k).

**Fig. 5 Ubiquitination of BARD1 by RNF19A is essential for HR regulation. a** HEK293T cells transfected with WT or NES deletion (ΔNES) mutant of Flag-BARD1 were lysed and pulled down by Ni-NTA and immunoblotted with indicated antibodies. **b** Indicated plasmids were transfected into HEK293T cells. Cell lysates were pulled down by Ni-NTA and immunoblotted as indicated. **c** WT or 2KR Flag-BARD1 were transfected into HEK293T cells. Cell lysates were immunoprecipitated with Flag beads and subjected to immunoblot with indicated antibodies. **d** Indicated plasmids were transfected into HEK293T cells. After 6 h MG132 treating, cell lysates were immunoprecipitated with Flag beads and subjected to immunoblot with indicated antibodies. **e**–**g** U2OS cells stably expressing Vec or RNF19A (exo) using plvx2-CMV-RNF19A (no tag) were further stably overexpressed with WT or 2KR Flag-BARD1. Each group of cells was transfected with BARD1 siRNAs for 48 h and treated with IR (2 Gy, 1 h). BRCA1 and Flag-BARD1 foci were detected by immunofluorescence (**e**). Quantification of focus signals per cell ($n = 100$) is shown in **f**: Flag-BARD1 and **g** BRCA1. Scale bars, 10 μM. **h** HEK293T cells stably expressing Vec or RNF19A (exo) were further stably overexpressed with WT or 2KR Flag-BARD1 and transfected with BARD1 siRNAs for 48 h and subjected to DR-GFP based HR assay. **i** The sensitivity of cells shown in (**e**) to Olaparib was assessed using colony formation assay. **j** U2OS ER-*AsiSI* cells stably expressing Vec or GFP-RNF19A were further stably overexpressed with WT or 2KR Flag-BARD1 and transfected with BARD1 siRNAs for 48 h and treated with 4-OHT for 4 h. Cells were digested and measured of DNA resection. **k** U2OS cells stably expressing Vec or GFP-RNF19A were further stably overexpressed with WT or 2KR Flag-BARD1 and treated with IR (10 Gy, 1 h). Nuclear (Nuc) and cytoplasmic (Cyto) proteins were extracted respectively. Cell lysates were immunoblotted with indicated antibodies. Error bars represent means ± s.d. of three (**f**, **g**, **i**, and **j**) or four (**h**) independent experiments. $p$ values are determined by unpaired two-sided $t$ test in **f**–**j**. Source data are provided as a Source Data file.

Taken together, these results suggest that ubiquitination of BARD1 by RNF19A on K96 and K110 is important for regulating association between BARD1 and BRCA1, as well as their HR-mediated function at DSB sites.

**Role of RNF19A in tumorigenesis and cancer therapy.** RNF19A is amplified at the mRNA level in many human cancers, especially in breast cancer (BC) and ovarian carcinoma (Supplementary Fig. 6a, cBioportal dataset[48]). To investigate the clinical relevance of RNF19A in cancer, we first examined whether RNF19A impacts BC response to chemotherapy. We overexpressed RNF19A in MDA-MB-231 and HCC1806 BC cells, which express RNF19A at relatively low levels (Supplementary Fig. 6b). As shown in Fig. 6a, b, overexpression of RNF19A markedly sensitized cells to Olaparib. We then subcutaneously implanted MDA-MB-231 cells to further confirm the effect of RNF19A to Olaparib in xenograft models. Overexpression of RNF19A slightly promoted cancer cell growth without drug intervention while mice bearing RNF19A overexpressing MDA-MB-231 cells displayed more noticeable tumor shrinkage in the Olaparib treatment group (Fig. 6c–e). These results suggest high expression of RNF19A renders cancer cells hypersensitivity to PARPi.

To evaluate the clinical importance of RNF19A in BC progression, the correlation between BARD1/ RNF19A expression levels and clinical features was analyzed through human BBC tissue microarray (TMA) with 140 specimens, containing 46 pairs of matched breast tumor tissue and corresponding adjacent non-tumor breast tissue samples (Supplementary Table 1). The difference in RNF19A staining between tumor and adjacent tissues was statistically significant (Supplementary Fig. 6c, d), consisting of the mRNA expression results from our database search. RNF19A protein level was overexpressed in more than half (79/140, 56.4%) of the BC tissues and this group of patients exhibited improved overall survival (OS) (Supplementary Fig. 6e), while only 23.9% (11/46) of the non-BC tissues were stained positively for RNF19A.

Studies have indicated that cancer cells overexpressing BARD1 are resistant to DNA-damaging chemotherapy and radiotherapy, thus BARD1 overexpression is associated with poor prognosis in cancer patients[49]. Our analysis supported a similar trend in 69 samples with an upregulation of BARD1 showing a remarkable reduction in OS compared with 71 samples scoring as low expression (Fig. 6f). Next, we proceeded to investigate whether the correlation between BARD1 level and prognosis would be affected by RNF19A expression. As shown in Fig. 6g, in a subgroup with high expression of BARD1, more than half of the cases had accompanied RNF19A upregulation and better OS. As for patients with relatively low BARD1 levels, the status of

RNF19A didn't further influence clinical outcomes (Fig. 6h). These results are consistent with our findings that RNF19A suppresses BARD1's function on HR, thus reversing the resistance to chemo- and radiotherapy caused by BARD1. Accordingly, the influence of RNF19A on the prognosis of BC patients is BARD1-dependent. On the other hand, RNF19A's repression of BARD1-dependent HR function might precipitate normal cell transformation. Since RNF19A regulates the nuclear export of BARD1, and previous studies reported a significant increase of BARD1 level in the cytoplasm of many cancer cells[50], we assessed BARD1 distribution in the nucleus and cytoplasm. BARD1 was localized in both the cytoplasm and nucleus in normal tissues; and, both compartments underwent distinct cancer-related changes (Supplementary Fig. 6f, g). Compared to adjacent tissue, there is a decreased proportion of nuclear/cytoplasmic BARD1 in tumor samples (Fig. 6i and Supplementary Fig. 6g), indicating the specific upregulation of cytoplasmic BARD1 in BC tissues. Moreover, a negative correlation between increased RNF19A and decreased percentage of nuclear BARD1 was observed in tumor tissues (Fig. 6j, k). Collectively, our findings suggest that overexpression of RNF19A contributes to decreased nuclear BARD1 in BC tissues and confers a better prognosis for patients.

## Discussion
The BRCA1/BARD1 heterodimer complex is essential to promote high-fidelity HR repair for DSBs. Dissociation of the heterodimer due to various factors will result in instability of both proteins, profoundly affecting their roles in promoting DNA repair and maintaining genome stability. Intriguingly, BRCA1/BARD1 can block nuclear export of each other by covering their own NES, both located in their binding domains, resulting in their nuclear anchorage and enhanced function. Here we identified that the RBR E3 ligase RNF19A, is a fine-tuning protein involved in this process. Mechanistically, RNF19A compromises BRCA1/BARD1 association by a ubiquitin-dependent unmasking of their NES region, thereby inhibiting HR process (Fig. 7). We demonstrate that RNF19A is a critical factor that restrains BRCA1/BARD1 function. Clinically, high expression of RNF19A is associated with improved prognosis in BC patients with high levels of BARD1. Thus, the characterization of RNF19A as an E3 ligase for BARD1 elucidates the dynamic regulation of BRCA1/BARD1 complex formation for HR repair.

Previous studies showed that NES1 (aa 92–111) and NES2 (aa 102–120) are the two main sequences bearing nuclear export activity in BARD1[20]. However, whether and how the nuclear export of BARD1 is regulated at the physiological level is unclear. Mutation in the RING domain of BRCA1 or BARD (for example, C61G of BRCA1 and L107A of BARD1) could potentially affect

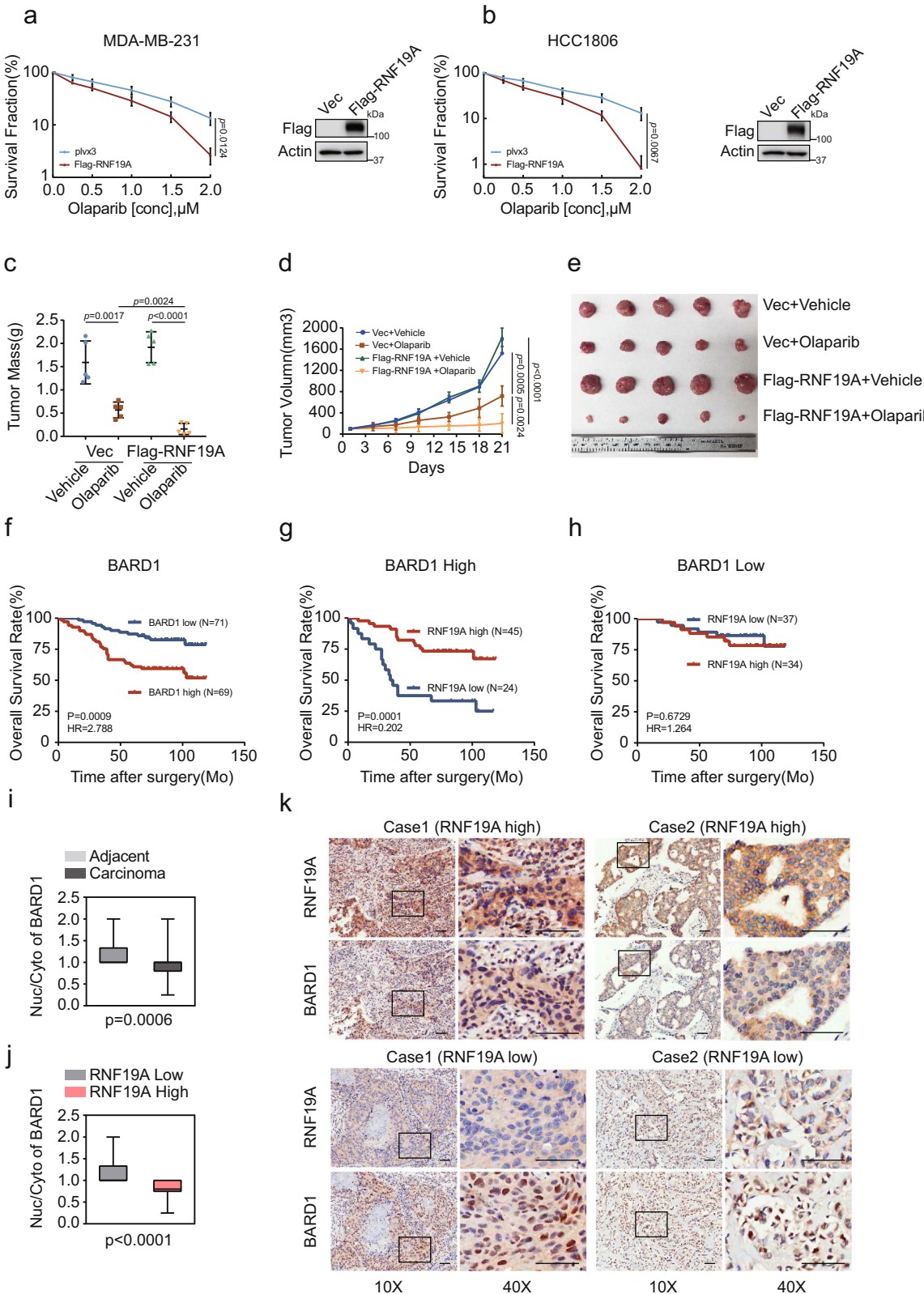

their interaction and nuclear export[7,51]. Intriguingly, we found two lysine sites within the NES1 (K96) and NES2 (K110) as ubiquitinated sites targeted by RNF19A. Furthermore, ubiquitination within the NES region of BARD1 inhibits its connection with BRCA1, which then determines the fate of HR process. To our knowledge, except for BRCA1/BARD1 autoubiquitination, it is another identification of an E3 ubiquitin ligase for BARD1. The

over-suppression of BRCA1/BARD1 function by RNF19A-mediated ubiquitination might cause genomic instability and promote tumorigenesis. Indeed, we observed a negative correlation between RNF19A expression and nuclear/cytoplasm ratio of BARD1, which is lower in tumor tissues compared with adjacent tissues. On the other hand, studies have implicated that hyper-activation of HR is also responsible for genome-

**Fig. 6 The role of RNF19A in response to cancer therapy. a–b** Sensitivity of control and RNF19A overexpression MDA-MB-231 (**a**) and HCC1806 cells to Olaparib was assessed by colony formation assay. **c–e** MDA-MB-231 cells stably expressing Vec or Flag-RNF19A were subcutaneously injected into nude mice. Tumor weight (**c**) and volume (**d**) were measured as indicated. Tumor images were acquired as shown in **e**. **f–h** Kaplan–Meier estimates of overall survival of breast cancer patients with different expression levels of BARD1 and RNF19A. The log-rank test was used to compare the survival curves between groups. **i** Nuclear (Nuc) cytoplasmic (Cyto) BARD1 staining was evaluated by the German semi-quantitative scoring system according to the staining intensity and the proportion was compared between tumor ($n = 140$, mini-0.25, max-2, centre-1, lower quartile-0.8, upper quartile-1) and adjacent tissues ($n = 46$, mini-1, max-2, centre-1, lower quartile-1, upper quartile-1.33). **j** Proportion of nuclear (Nuc)/cytoplasm (Cyto) staining of BARD1 was compared between RNF19A high ($n = 79$, mini-0.25, max-1, centre-0.8, lower quartile-0.75, upper quartile-1) and low ($n = 61$, mini-1, max-2, centre-1, lower quartile-1, upper quartile-1.33) expression subgroups. **k** Representative images of IHC analysis of RNF19A and BARD1 in the serial sections of tumor tissues. Scale bars, 100 μM. Values are means ± s.d. of three (**a** and **b**) or five (**c** and **d**) independent experiments. *p* values are determined by unpaired two-sided *t* test in **a–d** and **i–j**. Source data are provided as a Source Data file.

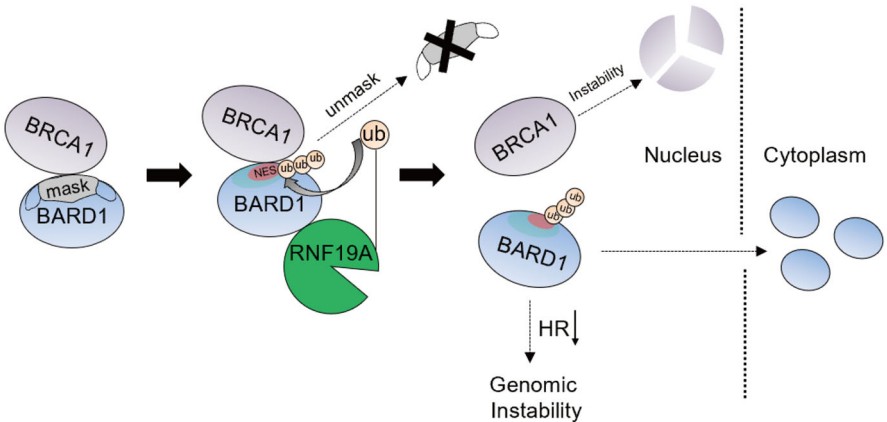

**Fig. 7 Schematic model of BARD1 regulation by RNF19A.** RNF19A interacts with BARD1 and promotes its ubiquitination. The ubiquitination of BARD1 unveils the NES of BARD1 located within its RING domain, resulting in the dissociation of BRCA1/BARD1 complex and the export of BARD1 to the cytoplasm.

destabilizing events, owing to the fact that the repair templates from sister chromatids utilized by HR are often not perfectly homologous[52]. In such a context, the negative regulation of BARD1 by RNF19A might be important for maintaining genomic stability. Thus, RNF19A acts as a fine-tuning coordinator in BRCA1/BARD1 complex might be essential to mediate the balance of HR repair.

Multiple studies strongly support that HR-based DNA repair affects clinical outcomes of cancer treatment and drug resistance. A compelling example is the application of PARPi and platinum-based agents in BRCA1/2 mutant ovarian and BC[53,54]. On the other hand, high expression of BRCA1 or BARD1 is associated with a poor prognosis. Targeting the BRCA1/BARD1 complex may provide a promising strategy for those tumors in cancer therapy. Our results imply that overexpression of RNF19A sensitizes BC cells to PARPi as well as radiotherapy. Recently, a large genome-scale CRISPR screen against DNA-damaging agents scored RNF19A as one of the potential Olaparib-sensitizing hits[55]. Levels of RNF19A together with BARD1 expression predict the prognosis of patients to varying degrees.

Taken together, we present evidence that RNF19A acts as an E3 ligase for BARD1 and demonstrate how RNF19A is involved in BRCA1/BARD1-mediated HR repair. Moreover, we report that expression levels of RNF19A together with BARD1 might provide prognostic guidance for BC patients. We propose a model where ubiquitination of BARD1 influences its distribution and association with BRCA1 to affect HR process, genomic stability, and chemotherapy response. Thus, our results identify RNF19A as a DNA repair-related factor and provide insights into the mechanism of therapy response for BC patients. These findings might also be extended to other cancer types such as prostate cancer and ovarian cancer.

## Methods

**Cell culture, plasmids, reagents, and antibodies.** HEK293T, MDA-MB-231, U2OS, and HCC1806 cells were purchased from ATCC. HEK293T, MDA-MB-231 were cultured in Dulbecco's Modified Eagle Medium (DMEM) with 10% fetal bovine serum (FBS). U2OS and HCC1806 cells were cultured in McCoy's 5 A and RPMI1640 with 10% FBS, respectively. All cell lines have been identified by the medical genome facility center of Mayo Clinic (Rochester, Minnesota). ER-*AsiSI* U2OS cells were obtained from Dr. Gaëlle Legube's lab (University of Toulouse, France) and cultured in McCoy's 5 A with 10% FBS. Clz3 cells were kindly provided by Dr. Zhiyong Mao (Tongji University, China) and cultured in DMEM with 10% FBS.

RNF19A-KO U2OS cells were generated via CRISPR/Cas9, using Lenti-CRISPR V2 containing a gRNA targeted RNF19A (CAAGCTCACAGATGAAGCGA). RNF19A-KO cells were maintained in McCoy's 5 A with 10% FBS.

**Plasmids and antibodies.** RNF19A shRNA (#1 5′-CGCAAGATTCACAATCGC TAT-3′, targeting CDS; #2 5′-GCCGGGTTTCATTATCATAAT-3′, targeting 3′-U TR), BARD1 shRNA (5′-TGAAAGTATGAAATCGCTATT-3′, targeting CDS), RA D51 shRNA (5′-CGGTCAGAGATCATACAGATT-3′, targeting CDS)) and XRC C4 shRNA (5′-CCTCAGGAGAATCAGCTTCAA-3′, targeting CDS) were purchased from Sigma.

RNF19A full-length cDNA was purchased from Open Biosystem and cloned into the following vectors for mammalian expression: pLVX2-CMV-puro (no epitope tag), pLVX3-CMV-puro (3xFlag tag at N-terminus), and pLVX6-CMV-puro (GFP tag at N-terminus). pcDNA5-FRT-HA-BARD1 was a gift from Dr. Parvin Jeffrey and subcloned into PLVX3/6 -CMV-puro lentiviral plasmids. Flag-BARD1 truncations with BRCT domain deletion were gifts from Dr. Huadong Pei. All point-mutations and truncations were performed by a PCR-based site-directed mutagenesis method and confirmed by Sanger sequencing. Cloning primers are summarized in Supplementary Table 2. Control and ON-TARGET plus BARD1 (#L003873) and 53BP1 (#L003548) siRNAs (Smartpool) were purchased from Dharmacon and were transfected with TransIT-X2 transfection reagent according to the manufacturer's instructions. HA-tagged ubiquitin and ubiquitin lysine mutants, HA-BRCA1 were obtained from Addgene.

Antibodies used in this study were as follows: anti-RNF19A (ab251750, WB-1:500, IHC-1:200) was purchased from Abcam. Anti-BARD1 (GTX132094, WB-1:1000, IF-1:1000, IHC-1:500) and anti-RAD51 (GTX100469, WB-1:1000, IF-1:1000) were purchased from Gentex. Anti-BRCA1 (sc-6954 WB-1:1000, IF-1:1000), anti-Ub (sc-8017, WB-1:1000), anti-GFP (sc-9996, WB-1:1000), anti-CtIP

(sc-271339 WB-1:1000) and anti-RPA3232 (sc-56770, IF-1:1000) were obtained from Santa Cruz. Anti-γ-H2AX (05-636, IF-1:1000), anti-FK2 (04-263, IF-1:1000) and anti-MDC1 (05-1572, IF-1:1000) were purchase from Millipore. Anti-CtIP (61141 IF-1:1000) was purchased from Active Motif. Anti-53BP1 (NB100-304, WB-1:1000, IF-1:1000) was from Novus Biologicals. Mouse and rabbit anti-FLAG (F1804 and F7425, WB-1:3000, IF-1:1000), anti-HA (H9658, WB-1:3000), and anti-β-actin (A2228, WB-1:3000) antibodies were purchased from Sigma. Normal rabbit IgG (12-370) and mouse IgG (12-371) were purchased from Sigma. Alexa Fluor 488-labeled Donkey anti-Rabbit IgG (H + L, 715-585-150,1:3000), Alexa Fluor 594-labeled Donkey anti-Rabbit IgG (H + L, 711-585-152, 1:3000), Alexa Fluor 488-labeled Donkey anti-Mouse IgG (H + L, 715-545-150, 1:3000), Alexa Fluor 594-labeled Donkey anti-Mouse IgG (H + L, 715-585-150, 1:3000), Donkey anti-Mouse IgG (H + L,715-675-151, 1:3000) and Donkey anti-Rabbit IgG (H + L,711-675-152, 1:3000) were purchased from Jackson ImmunoResearch.

**Lentivirus packaging and infection.** Lentiviruses for infection of HEK293T, MDA-MB-231, HCC1806, and U2OS were packaged in HEK293T cells using TransIT-X2 transfection reagent. In all, 48 h after transfection, the medium was collected and added to the target cells with 8 μg/ml polybrene to enhance infection efficiency.

**Immunofluorescence.** Cells were cultured on coverslips 24 h before experiments. For γ-H2AX, MDC1,53BP1, FK2, BRCA1, and BARD1 foci, cells were fixed with 4 % paraformaldehyde (PFA), permeabilized with 0.5% Triton X-100; For RAD51 foci, cells were permeabilized with 0.5% Triton X-100 on ice for 5 min, then fixed with 4% PFA; Cells were fixed and permeabilized with methanol: acetone (1:1) at −20 °C for 20 min to detect RPA32 foci. Cells were incubated with primary antibodies (4 °C overnight) and subsequently incubated with corresponding Alexa Fluor 488 or 594-conjugated secondary antibodies (37 °C, 20 min) and the nuclei were stained with DAPI. The coverslips were mounted onto glass slides using an anti-fade solution and visualized by a Nikon eclipse 80i fluorescence microscope and related software. Foci intensity was quantified using Image J.

**Immunoprecipitation (IP) and western blotting.** Cells were lysed (20 mM Tris-HCl, pH 8.0, 100 mM NaCl, 1 mM EDTA, 0.5% Nonidet P-40 with protease inhibitors) and centrifuged at 14,000 × g for 15 min. The supernatant was subjected to Flag M2/ HA beads (Sigma Aldrich) or indicated antibodies with protein A/G-Sepharose beads (Amersham Biosciences) and rotated overnight at 4 °C. Beads were washed with NETN buffer three times, and samples were boiled with 50 μl 1* sodium dodecyl sulfate (SDS) loading buffer and immunoblotted with indicated antibodies.

**Protein purification and in vitro pull-down assays.** Human RNF19A cDNA was cloned into the GST fusion vector pGEX-4T-1. His-BARD1 (26–327) and BRCA1(1–304) based on Pet28a and pET151D topo were purchased from addgene (#12646and #12645). E. coli (BL21 DE3 strain) transformed with GST- or His-tagged constructs were induced for 20 h at 18 °C with 0.4 mM IPTG or 4 h at 37 °C with 1 mM IPTG to express GST or His fusion proteins. GST fusion was affinity purified using Glutathione-Sepharose beads (Sigma) and bound protein was diluted with GSH elution Buffer. His-tagged proteins were affinity purified using Ni-NTA agarose (Qiagen) and bound proteins were eluted with elution buffer (25 mM Tris-HCl, 200 mM NaCl, 500 mM imidazole)[56]. Purified proteins were used for in vitro GST pull-down assays. In brief, GST-RNF19A proteins were immobilized to GST beads, which were then incubated with purified His-tagged proteins or His alone. After washing, the proteins bound to the beads were analyzed by western blot.

**Denaturing Ni-NTA pull-down.** Transiently transfected or virus-infected cells were harvested and pellets were washed once in PBS. Cells were lysed in 8 M Urea, 0.1 M NaH$_2$PO$_4$, 300 mM NaCl, 0.01 M Tris (pH 8.0). Lysates were sonicated to shear DNA and incubated with Ni-NTA agarose beads (QIAGEN) for 1 h at room temperature. Beads were washed 3–5 times with 8 M Urea, 0.1 M NaH$_2$PO$_4$, 300 mM NaCl, 0.01 M Tris (pH 8.0). Input and beads were boiled in loading buffer and subjected to SDS-PAGE and immunoblotting.

**Denaturing immunoprecipitation for ubiquitination.** Harvested cells were lysed in 100 μl 62.5 mM Tris-HCl (pH 6.8), 2% SDS,10% glycerol, 20 mM NEM, and 1 mM iodoacetamide, boiled for 15 mins, then diluted 10 times with NETN buffer containing protease inhibitors as well as 20 mM NEM and 1 mM iodoacetamide. After removing cell debris, the cell extracts were subjected to immunoprecipitation with indicated antibodies.

**In vitro ubiqtination assays.** The in vitro ubiquitination assay was performed in 50 μl reaction buffer (50 mM Tris, pH 7.4, 1 mM DTT, 2 mM ATP, 5 mM MgCl$_2$) with 0.5 μg E1 (Boston Biochen), 0.5 μg E2 (UbcH7; Boston Biochem), 5 μg Ub (Boston Biochen), 0.5 μg purified His-BARD1 protein and 0.5ug purified GST-RNF19A for 1 h at 37 °C. Samples were then subjected to western blot with anti-Ub antibody.

**HR, NHEJ, and SSA reporter assay.** Control and RNF19A knockdown Clz3 cells, which contain a dual reporter for HR-tdTomato and NHEJ-GFP, were treated with doxycycline for 48 h to turn on I-SceI expression and to induce DSBs. HEK293T Cells transfected with indicated plasmids, shRNAs or siRNAs were then transfected with pCBA-I-SceI, pCherry, and either DR-GFP or SSA-GFP. In all, 48 h after transfection, cells were harvested and fixed, the percentage of tdTomato/GFP positive cells were counted by flow cytometry (FACS). The graphical account for FACS sequential gating/sorting strategies was shown in Fig. 1g and Supplementary Fig. 7

**Sister chromatid exchange assay.** WT and RNF19A-KO U2OS cells were grown through two cell cycles (42 h) in the presence of 100 μM of BrdU after treating with or without DNA-damaging agents (IR 0.5–1 Gy). Colcemid (Sigma) was added to a final concentration of 0.2 μg/ml during the last 2 h of BrdU treatment to accumulate mitotic cells. Harvested cells were treated with 75 mM KCl hypotonic solution for 30 min at 37 °C and then fixed with 3:1 methanol/acetic acid. Fixed cells were dropped onto a glass slide and air-dried, stained with 5% Giemsa for 10 min after being heated at 88 °C for 10 min in 1 M NaH2PO4 buffer (pH 8.0), and visualized by fluorescence microscopy.

**DNA resection measurement.** ER-AsiSI U2OS cells expressing indicated shRNAs, constructs, or siRNAs were harvested after 1 μM 4-OHT treatment for 4 h. Genomic DNA was extracted using DNAzol reagent (Invitrogen) followed by the manufacturer's protocol. A total of 500 ng DNA was digested with mock or BsrGI restriction enzyme at 37 °C overnight. In all, 2 μl (~20 ng) digested or mock samples were used as templates in 25 μl of qPCR reaction (12.5 μl 2*Taqman Universal PCR Master Mix-ABI, 0.5 μM each primer and 0.2 μM probe). The proportion of ssDNA generated by resection at three selected sites was calculated as previously described[57].

**Quantitative real-time PCR (qRT-PCR).** mRNA was extracted with RNAiso Plus reagent (Takara). qRT-PCR was carried out with Power SYBR Green PCR Master Mix (Takara). Quantification of gene expression was calculated on the basis of the $2^{-\Delta\Delta CT}$ value normalized the GAPDH. Primers used for RT-PCR:
BARD1-F: 5′-GAGCCTGTGTTTAGGAGGA-3′;
BARD1-R: 5′-ACTTCGAGGGCTAAACCACA-3′;
GAPDH-F: 5′-CAGCCTCAAGATCATCAGCA -3′;
GAPDH-R: 5′-TGTGGTCATGAGTCCTTCCA -3′.

**Tumor xenograft.** Experiments were performed under the approval of the Institutional Animal Care and Use Committee at Mayo Clinic (Rochester, MN). MDA-MB-231 cells stably expressing plvx3 vector or Flag-RNF19A were injected subcutaneously into the flanks of 5-week-old female Balb/c athymic nude mice using 19-gauge needles. Each mouse received injections of a 200 μl mixture of 2 × 10$^6$ cells in PBS 2:1 with growth factor reduced Matrigel (BD Biosciences). Mice bearing tumors around 150 mm$^3$ were randomly divided into the vehicle (10% DMSO with 10% 2-hydroxypropyl-β-cyclodextrin daily) or Olaparib intervention (50 mg/kg daily) groups. Tumor volume was measured every 3 days using calipers when treatment began and calculated as length × width$^2$. After 3 weeks' treatment, mice were euthanized and tumors were dissected and weighted. The maximal tumor size has not exceeded the range permitted by the ethics committee (tumor size ≤2000 mm$^3$).

**Tissue samples, immunohistochemistry (IHC), and stratification of expression.** TMA chips containing 46 pairs of BC and adjacent tissues plus 94 BC tissue samples with related clinicopathological information were purchased from Shanghai OUTDO Biotech Co, Ltd (Shanghai).
　　IHC assays performing on TMA chips were subjected to a standard labeled streptavidin-biotin protocol (Dako, Carpinteria, CA, USA) with RNF19A (Abcam) or BARD1 (GeneTex) antibodies. RNF19A and BARD1 expression scores were assessed based on the German semi-quantitative scoring system according to the staining intensity and the staining region as previously described[58]. Both expression levels were dichotomized as low expression (score <4) and high expression (score ≥4) in tumor and adjacent tissues.
　　The diagnosis of normal tissue or BC was confirmed by independent pathologists based on histological findings. All experiments were performed with informed consent obtained from all subjects with the approval of the Medical Ethics Committee of Shanghai Jiao Tong University School of Medicine Review Board.

**Statistics and reproducibility.** Data in bar or line graphs are presented as means ± s.d. of at least three independent experiments. Western blotting and micrograph data were repeated independently three times with similar results. For the animal xenograft study, data are presented as means ± s.d. of five biologically independent samples. Statistical significance was analyzed by two-tailed unpaired Student's t test, log-rank test, and Pearson's χ$^2$ test in Microsoft Excel 2016 and GraphPad Prism 7. The flow cytometry data were collected using Attune NxT v2.6 and analyzed by flowjo V10.

**Reporting summary**. Further information on research design is available in the Nature Research Reporting Summary linked to this article.

## Data availability

All data are available from the authors upon reasonable request. Source data are provided with this paper.

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

## Acknowledgements

We thank Parvin D. Jeffrey (Ohio State University) and Huadong Pei (GW Cancer Center) for providing BARD1 constructs. We thank Dr. Gaëlle Legube's lab (University of Toulouse, France) for providing ER-*Asi*SI U2OS cells and Dr. Zhiyong Mao (Tongji University, China) for providing Clz3 cells. We thank the members of the Lou lab for comments and discussion throughout the project. This work was supported in part by grants from Mayo Foundation, Mayo Beast Cancer SPORE and Shanghai Science and Technology Committee (19411950900 to Chen J). J.H. was supported by the Mayo Edward C. Kendall Fellowship in Biochemistry Award. J.A.K. was supported by T32GM65841.

## Author contributions

Q.Z. performed most of the experiments. H.H. performed IHC analysis. H.L., P.Yi., Y.C., M.G., G.G., M.D., K.L., J.S., X.T., and Y.Z. provided technical and data analysis assistance. P. Yin provided reagents for experiments. Q.Z. and J.H. designed and interpreted the experiments and wrote the manuscript; J.A.K. participated in proofreading this manuscript; Q.Z., J.H., J.Y., J.C., and Z.L. revised the manuscript; Z.L. conceived and supervised the entire project.

## Competing interests

The authors declare no competing interests.
