## [Peer Review File · Nature Communications]

REVIEWER COMMENTS

Reviewer #1 (Remarks to the Author):

Manuscript by Zhu et al. identifies an RBR protein (ring between ring fingers) RNF19A that affects BARD1 nuclear localization. They claim that RNF19A does this by ubiquitinating BARD1 and forcing dissociation between BRCA1 and BARD1. This dissociation results in unveiling NES on BARD1 forcing it out of nucleus and into the cytoplasm. They go on to show that this removal of BARD1 from the nucleus results in HR downregulation. Given that tumor cells with defective HR are especially prone to killing by PARPi, they also make the case that RNF19A levels in tumors can dictate the response of these tumors to PARPi based therapy. The authors identify two residues on BARD1 (K96 and K110) that are ubiquitinated, and go on to show that ubiquitination at these residues results in loss of interaction between BRCA1 and BARD1 and reduced HR.

Overall, this manuscript provides some exciting results that identify RNF19A as a novel modulator of BARD1 ubiquitination. Importantly, through the tumor studies and analysis of breast tumor sections, it shows that RNF19A levels can affect tumor progression and/or response to therapy (Olaparib). However, the manuscript does not deliver on the main claim that RNF19A is an E3 ligase for BARD1. No direct analysis (in vitro ubiquitination study with purified proteins) is carried out to confirm this conclusion. This is an interesting study which will benefit from addressing the points/concerns listed below.

Major Points:

1. Fig. 1A and Fig. S1A - The difference in g-H2AX intensity in RNF19A depleted cells (Fig. 1A) or overexpressing cells (Fig. S1A) compared to the relevant control does not look very convincing. Furthermore, are the cells in Fig. S1A depleted of endogenous RNF19A? If not, then how much more is the overexpression compared to endogenous RNF19A levels?
2. Authors study recruitment of BRCA1/BARD1/RPA/RAD51 by looking at foci formation. Fig. 1I quantifies data in Fig. 1H by counting cells with >10 foci. However, after going through the images in Fig. 1H, it seems that there is not much difference in the number of foci and at the very least the data presented for controls, shows equal number of cells with >10 foci as shRNF19A? It is not clear how this quantification is done. Any why have different IR doses been used for different proteins (as per Fig. legend Fig. 1I)?
3. For HR/NHEJ assays, the system being used here is a transient transfection system instead of traditional stably integrated DR-GFP system. In this case how are the authors ensuring that there are no differences in transfection efficiency of these constructs which could contribute to HR differences that they see.

4. What is the effect of the hyper-recombination after RNF19A downregulation on chromosomal stability/aberrations? Is there also increased sister chromatid exchange?
5. When studying the effect of WT or CA mutant Flag-RNF19A on BRCA1/BARD1 recruitment to DSBs, the authors do not show evidence of equal DNA damage (for eg. gH2AX staining) in these over expressed cells.
6. It is not clear what is happening in Fig. 4C. The legend says that the lysate is immunoprecipitated with HA which should bring down all Ubiquitinated proteins given that HA-Ub is being used. But the Figure shows that the IP is with BARD1 (IP:BARD1)?
7. The only way to confirm that RNF19A is 'directly' ubiquitinating BARD1 is to work with recombinant proteins. Without such an experimental strategy, there is no way to rule out that ubiquitination of BARD1 is instead because of another intermediate protein/s that might be getting ubiquitinated by RNF19A.
8. Why is that there is equal amount of BRCA1 in G1 and S phase (Fig. S4D)? It is well established that BRCA1 is most expressed in S phase. Furthermore, it is also not clear why in Input section, there is no BRCA1 in lane 1 (IgG; DTB 0hr)? If this is Input, at the very least it should have same amount of BRCA1 in lane 1 and lane 2 (both are DTB 0hr).

Minor Points:

1. Lines 85-87 needs to re-worded. That section is repetitive and seems to be missing some information.
2. It might be easier to address the effect, if any, of RNF19A loss by studying this in RNF19A CRISPR knockout cells instead of siRNA treated cells.
3. Figure 2F – what is the effect of other RNF19A mutants (eg. R2 and R3) on HR efficiency? Given that they interact with BARD1, do they not have an effect on HR?
4. Typo in Figure S3A – should be shRNF19A instead of shRNA19A
5. Line 323 should say “consistent” instead of “consisting”.
6. The effect of RNF19A overexpression on BRCA1 stability will be better addressed by using MG132 to block protein degradation.

Reviewer #2 (Remarks to the Author):

In the manuscript titled “Ubiquitinated unmasking of BARD1 by RNF19A prevents BRCA1/BARD1 dependent homologous recombination”, the authors described the functional and mechanistic characterization of a newly discovered ring between ring (RBR) protein RNF19A. The authors showed that RNF169A is a suppressor for homologous recombination repair by negatively regulating BARD1. It is convincing that RNF19A plays an important role in the DDR. Additionally, this study provided very interesting mechanistic regulation on RNF19A-mediated BARD1 ubiquitination and its functional implications in HR repair. Moreover, the authors revealed the connection between RNF19A expression and triple-negative breast cancer, which potentially provides a therapeutic target for cancer treatment. In general, the study is well-designed and the findings presented in the manuscript are very novel. The data on the molecular and cell biology are particularly solid. Overall, this is an important discovery. The study is likely to be of considerable interest to the DNA repair and cancer etiology community. However, there are a few concerns that are needed to be addressed and clarified.

Major comments:

1) The current manuscript did not show the RNF19A localization or whether DNA damage alters the RNF19A subcellular localization or chromatin loading? Also, is RNF19A recruited to DNA double-strand breaks? If so, what is the upstream genetic pathway?

2) The authors presented here in the manuscript that RNF19A has two RING domains. However, it is unclear which RING domain confers the catalytic activity. Is there a rationale behind mutating the specific RING domain in this study? There is also a discrepancy in the domain nomenclature compare to NCBI. I assume that the RING2 is referring to the IBR, half RING-finger domain. Comparing the sequences between the two RING domains may provide more information on their structure. Also, in vitro ubiquitination experiments should be included to support the in vivo finding.

3) The protein-protein interaction experiments were performed in cells. It is unclear if they interact directly. In vitro pull-down assay should be included to show whether the RING domain alone is sufficient for their interaction.

4) The authors showed that GFP-RNF19A overexpression impairs the BRCA1-BARD1 binding. Does RNF19A depletion enhance BARD1-BRCA1 interaction?

5) Since RNF19A is important in regulating the BRCA1-BARD1. It would be interesting to see the RNF19A expression profile throughout the cell cycle and whether the RNF19A-mediated BRCA1-BARD1 function is cell cycle-dependent?

Minor comments:

1) Some of the labels are inconsistent between text and figures, such as XRCC vs XRCC4; RPA vs RPA32? It could be confusing to the general reader. Also for ubiquitination vs ubiquitylation, even though they are interchangeable.

2) A precise detailed information of the deletion used in the study should be provided.

3) Figure 6A labels are off.

4) It would be helpful to provide a little bit more information for the RNF19A background in the revised manuscript.

Reviewer #3 (Remarks to the Author):

In this report, the authors proposed a model in which RNF19A may play a role to suppress HR activity for the fine-tuning of DSB repair. The authors showed that depletion of RNF19A enhances the proportion of DSBs undergoing end resection, which directs the repair pathway toward HR. Furthermore, the authors demonstrated that BARD1 is ubiquitylated by RNF19A, and that this ubiquitylation step is associated with BRCA1 stability and BARD1 nuclear export; hence, the authors proposed a model in which RNF19A suppresses HR at the step of resection by ubiquitylating BARD1. Some of their findings are interesting; however, several data are technically preliminary. More importantly, I am not convinced by the interpretation proposed by these authors, as follows. 1) The authors stated that RNF19A suppresses HR and that its depletion enhances HR; why does RNF19A need to suppress the accurate HR, i.e., what is the advantage of HR suppression for cells? If the

restored HR pathway is still accurate under RNF19A deficiency, cells do not require RNF19A. 2) The authors reported that the enhanced resection observed in RNF19A-depleted cells is dependent on BARD1; however, what is the mechanism underlying the BARD1-dependent resection? Although the authors referred to references 33 and 34, I do not think that these studies demonstrate the mechanism underlying the manner in which BARD1 promotes resection. Is it via the activation of the initiation of resection by CtIP/MRN or via the removal of the 53BP1 barrier by BRCA1? Overall, this conclusion remains preliminary and further investigation is required to elucidate this issue.

1. Major technical issue: the data presented in Figure 1 are confusing. The authors showed a faster DSB repair in cells treated with the RNF19A siRNA at 8 h after IR. I assume that the authors aimed to state that the enhanced HR activity accelerates DSB repair. However, this is unlikely to occur because HR is a slower repair pathway compared with NHEJ, i.e., the switch of repair pathway from NHEJ to HR delays the speed of the DSB repair kinetics. Hence, generally, the increase in the fraction of DSBs undergoing HR yields greater gH2AX foci compared with control cells. In the reporter assays depicted in Figure 1F-G, strangely, RNF19A depletion did not change NHEJ, even though HR was enhanced. When the HR pathway is enhanced, some other pathways should be downregulated. Which repair fraction is converted into HR? In normal human cells, the majority of DSBs are repaired by either NHEJ or HR. SSA has a minor impact on the DSB fraction in normal cells. Thus, I am not convinced by these data. Last, why did the authors use different time points and IR doses to examine DDR foci in Figure 1I? In the representative image, the authors showed >20-30 RPA/RAD51 foci (as far as I can see) at 3-5 h after 2 Gy. The number of RAD51 foci in cells treated with the RNF19A siRNA is clearly greater than that in control cells in this figure. However, in Figure 1A, the percentage of RNF19A-siRNA-treated cells with >10 gH2AX foci per cell at 8 h is ~10%-20%. I do not understand how the RAD51 foci are quickly repaired within 3 h (5-8 h post IR). The authors should perform the time-course experiment and show the number of foci, rather than the percentage of foci, as shown in other figures.

2. Depletion of HR blockers, such as RAP80, increases resection, which leads to an abnormal repair pathway (PMID: 21406551). I am surprised that the increase in unscheduled resection in RNF19A-depleted cells still conferred normal HR. To clarify the quality of HR and resection-mediated repair in RNF19A-depleted cells, the authors should examine SCEs and SSA.

3. The authors analyzed the cell-cycle distribution of RNF19A-depleted cells, stating that the depletion did not affect the cell-cycle status. However, in Figure 3A, B, the authors showed that almost all cells harbored RPA/RAD51 foci in RNF19A depleted cells. Does this mean that the RNF19A siRNA causes HR in G1 cells as well? The authors should perform an analysis of the foci using appropriate cell-cycle markers, such as CENPF/Cyclin A and BrdU/EdU, and should report the number of foci at each phase of the cell cycle.

4. Depletion of BARD1 causes BRCA1 degradation. In general, it is known that the presence of BRCA1 contributes to the stabilization of BRCA1-complex proteins, including CtIP. Thus, the authors should determine the level of the CtIP protein in BARD1-depleted cells.

5. In Figure 3, the authors stated that the enhanced resection detected in RNF19A-depleted cells is dependent on BARD1. To clarify whether it is dependent on the BRCA1/53BP1 axis (PMID: 27239795), the authors should investigate whether 53BP1 depletion restores resection in the BARD1-depleted background.

6. In Figure 4, the authors showed that RNF19A ubiquitylates BARD1. Is this ubiquitylation induced after DNA damage? Or is it independent of DNA damage?

7. In lines 250-251, the authors stated that “RNF19A is involved in nuclear-to-cytoplasmic shuttling of BARD1 in response to DNA damage”. The enrichment of nuclear BARD1 in immunoblotting was marginal after IR (lanes 1 and 2 in Figure S5C). Is this enrichment cancelled by the RNF19A siRNA? The authors should also show the enrichment of nuclear BARD1 in response to DNA damage by immunofluorescence, i.e. before and after IR, which should be feasible because the authors used immunofluorescence in other experiments.

8. In lines 251-253, the authors mentioned that “Previous studies have demonstrated that BRCA1 protein levels increase in late G1 and reach a maximum in S phase, while BARD1 is expressed at parallel levels throughout the cell cycle.” However, I could not detect an obvious alteration in the level of the BRCA1 protein level between the G1 and S/G2 phases under DTB-dependent cell-cycle synchronization. To demonstrate that cells are successfully synchronized, the percentage of the cell-cycle phase should be shown at each time point.

9. In the discussion, the authors stated that “The fine-tuning of BRCA1/BARD1 function by RNF19A-mediated ubiquitination might cause genomic instability and promote tumorigenesis.” I wonder whether this should be “The lack of fine tuning of BRCA1/BARD1 function...”. The authors should clarify whether they discuss the situation in control cells, in RNF19A-overexpressing cells, or in tumor cells.

We thank the reviewers for the detailed and constructive comments. All the suggestions have helped us greatly improve our manuscript.

Here is the detailed response to reviews' comments.

Reviewer #1 (Remarks to the Author):

Manuscript by Zhu et al. identifies an RBR protein (ring between ring fingers) RNF19A that affects BARD1 nuclear localization. They claim that RNF19A does this by ubiquitinating BARD1 and forcing dissociation between BRCA1 and BARD1. This dissociation results in unveiling NES on BARD1 forcing it out of nucleus and into the cytoplasm. They go on to show that this removal of BARD1 from the nucleus results in HR downregulation. Given that tumor cells with defective HR are especially prone to killing by PARPi, they also make the case that RNF19A levels in tumors can dictate the response of these tumors to PARPi based therapy. The authors identify two residues on BARD1 (K96 and K110) that are ubiquitinated, and go on to show that ubiquitination at these residues results in loss of interaction between BRCA1 and BARD1 and reduced HR.

Overall, this manuscript provides some exciting results that identify RNF19A as a novel modulator of BARD1 ubiquitination. Importantly, through the tumor studies and analysis of breast tumor sections, it shows that RNF19A levels can affect tumor progression and/or response to therapy (Olaparib). However, the manuscript does not deliver on the main claim that RNF19A is an E3 ligase for BARD1. No direct analysis (in vitro ubiquitination study with purified proteins) is carried out to confirm this conclusion. This is an interesting study which will benefit from addressing the points/concerns listed below.

Major Points:

1. Fig. 1A and Fig. S1A - The difference in g-H2AX intensity in RNF19A depleted cells (Fig. 1A) or overexpressing cells (Fig. S1A) compared to the relevant control does not look very convincing.

In the revised manuscript, we provided better quality representative images in Fig.1a and Supplementary Fig.1a, and recalculated the raw numbers of foci per cell for foci quantification as shown in Fig.1b and Supplementary Fig.1b.

Furthermore, are the cells in Fig. S1A depleted of endogenous RNF19A? If not, then how much more is the overexpression compared to endogenous RNF19A levels?

In supplementary Fig.1e, we utilized anti-RNF19A antibody to present the RNF19A protein level in cells (Supplementary Fig.1e).

2. Authors study recruitment of BRCA1/BARD1/RPA/RAD51 by looking at foci formation. Fig. 1I quantifies data in Fig. 1H by counting cells with >10 foci. However, after going through the images in Fig. 1H, it seems that there is not much difference in the number of foci and at the very least the data presented for controls, shows equal number of cells with >10 foci as shRNF19A? It is not clear how this quantification legend Fig. 1I)?

In the revised manuscript, we recalculated the total numbers of foci per cell for foci quantification. As shown in Figure 1i, depletion of RNF19A increased foci formation of BRCA1, BARD1, RPA and RAD51 (lane 4-6).

3. For HR/NHEJ assays, the system being used here is a transient transfection system instead of traditional stably integrated DR-GFP system. In this case how are the authors ensuring that there are no differences in transfection efficiency of these constructs which could contribute to HR differences that they see.

Thanks for pointing this out. In our experiment, pCherry was co-transfected as a control for transfection efficiency. The HR/NHEJ efficiency was calculated by the percentage of GFP+ cells among pCherry+ cells, which was also used by other researchers¹⁻³. To further confirm this result, we employed CLZ3 cell line, stably expressing a dual-reporter, for the simultaneous measurement of both HR and NHEJ efficiency at the same chromosomal site⁴. In this system, we detected an increased HR (tdTomato+) efficiency in RNF19A knockdown cells, while the NHEJ (GFP+) efficiency was mildly compromised (Fig. 1f and g).

These HR results are consistent in both assays, suggesting that RNF19A plays an important role in regulating HR.

4. What is the effect of the hyper-recombination after RNF19A downregulation on chromosomal stability/aberrations? Is there also increased sister chromatid exchange?

To answer reviewer's question, we performed the mitotic spread assay by comparing control with RNF19A knockout U2OS cells. RNF19A-depleted cells showed decreased chromosomal breaks compared with control cells (Supplementary Fig.1c and d), consistent with the results that loss of RNF19A promote HR repair. Furthermore, the frequency of sister chromatid exchange (SCE) was examined in control and RNF19A knockout (KO) U2OS cells treating with IR. As shown in Supplementary Fig. S3p and q, RNF19A-depleted cells demonstrated an increased frequency of SCE, which indicates the enhancing HR activity. All these data suggest that RNF19A is a negative regulator of HR.

5. When studying the effect of WT or CA mutant Flag-RNF19A on BRCA1/BARD1 recruitment to DSBs, the authors do not show evidence of equal DNA damage (for eg. γ H2AX staining) in these over expressed cells.

Thanks for the suggestion. In the revised manuscript, we examined the amount of DNA damage by γ -H2AX staining in U2OS cells reconstituted with RNF19A WT or CA. As shown in Supplementary Fig. 3h and i, the initial induction of γ -H2AX was equal in those cells, suggesting equal DNA damage in these cells.

6. It is not clear what is happening in Fig. 4C. The legend says that the lysate is immunoprecipitated with HA which should bring down all Ubiquitinated proteins given that HA-Ub is being used. But the Figure shows that the IP is with BARD1 (IP:BARD1)?

Thanks for pointing out this error. It should be HA IP. In the revised manuscript, we've corrected the label.

4d

7. The only way to confirm that RNF19A is ‘directly’ ubiquitinating BARD1 is to work with recombinant proteins. Without such an experimental strategy, there is no way to rule out that ubiquitination of BARD1 is instead because of another intermediate protein/s that might be getting ubiquitinated by RNF19A.

The ubiquitination of BARD1 by RNF19A was confirmed by *in vitro* ubiquitination assays using purified GST-RNF19A and His-BARD1(aa26-327) proteins. His-BARD1 (aa26-327) contains the RING domain, which was widely used by other researchers for BARD1 study^{5,6}. The two lysine sites-K96 and K110 of BARD1 targeted by RNF19A are in this fragment. As shown in Fig. 4c, WT GST-RNF19A promoted His-BARD1 ubiquitination in the presence of recombinant E1, E2 (UbcH7), ubiquitin proteins *in vitro*. The C316A mutation markedly reduced His-BARD1 ubiquitination *in vitro*.

4c

8. Why is that there is equal amount of BRCA1 in G1 and S phase (Fig. S4D)? It is well

established that BRCA1 is most expressed in S phase. Furthermore, it is also not clear why in Input section, there is no BRCA1 in lane 1 (IgG; DTB 0hr)? If this is Input, at the very least it should have same amount of BRCA1 in lane 1 and lane 2 (both are DTB 0hr).

We repeated the experiment by using anti-BRCA1 antibody for endogenous IP. BRCA1 expression was increased in S phase. Overexpression of RNF19A decreased the interaction between BRCA1 and BARD1 in S phase. Cell cycle synchronization was through releasing cells from double thymidine (Supplementary Fig.4e).

S4e

Minor Points:

1. Lines 85-87 needs to re-worded. That section is repetitive and seems to be missing some information.

Thank you for pointing out the wrong sentence. We've re-worded the sentence in the revised manuscript.

2. It might be easier to address the effect, if any, of loss by studying this in RNF19A CRISPR knockout cells instead of siRNA treated cells.

Thanks for the suggestion. In the revised manuscript, we generated RNF19A knockout U2OS cells using Lenti-CRISPR V2 containing a gRNA targeted RNF19A, and performed the mitotic spread assay by comparing control with RNF19A KO U2OS cells. The results are shown in #4 of major points.

3. Figure 2F – what is the effect of other RNF19A mutants (eg. R2 and R3) on HR efficiency? Given that they interact with BARD1, do they not have an effect on HR?

In the revised manuscript, we investigated the HR efficiency of RNF19A-R2/R3. As shown in Supplementary Fig.3b, RNF19A R3, which abolished the catalytic domain, couldn't re-suppress the enhanced HR efficiency induced by RNF19A deficiency as RNF19A WT, indicating the catalytic activity of RNF19A is critical for its regulation

of HR. RNF19A R2 (deletion of the IBR) could partially reverse the increase in HR repair caused by RNF19A deficiency (Supplementary Fig.3j). Previous studies showed the structure of IBR domain is similar to RING2 and it might be involved in the transportation of E2~Ub. Since the IBR domain is not necessary for the BARD1/RNF19A interaction, we assumed that although RNF19A's IBR domain doesn't contain an active site Cys, it plays a role in facilitating the E3 ligase activity of RING2, whose deletion will restrict the enzymatic activity of RNF19A and have a mild impact on HR.

4. Typo in Figure S3A – should be shRNF19A instead of shRNA19A

Thank you for pointing out this mistake. This label has been corrected.

5. Line 323 should say “consistent” instead of “consisting”.

Thank you for pointing out this mistake. This word has been corrected.

6. The effect of RNF19A overexpression on BRCA1 stability will be better addressed by using MG132 to block protein degradation.

Thanks for the constructive suggestion. We repeat the experiments in Supplementary Fig. 4d and Fig.5d by adding MG132.

Reviewer #2 (Remarks to the Author):

In the manuscript titled “Ubiquitinated unmasking of BARD1 by RNF19A prevents BRCA1/BARD1 dependent homologous recombination”, the authors described the functional and mechanistic characterization of a newly discovered ring between ring (RBR) protein RNF19A. The authors showed that RNF19A is a suppressor for homologous recombination repair by negatively regulating BARD1. It is convincing that RNF19A plays an important role in the DDR. Additionally, this study provided very interesting mechanistic regulation on RNF19A-mediated BARD1 ubiquitination and its functional implications in HR repair. Moreover, the authors revealed the connection between RNF19A expression and triple-negative breast cancer, which potentially provides a therapeutic target for cancer treatment. In general, the study is well-designed and the findings presented in the manuscript are very novel. The data on the molecular and cell biology are particularly solid. Overall, this is an important discovery. The study is likely to be of considerable interest to the DNA repair and cancer etiology community. However, there are a few concerns that are needed to be addressed and clarified.

Major comments:

1) The current manuscript did not show the RNF19A localization or whether DNA damage alters the RNF19A subcellular localization or chromatin loading? Also, is RNF19A recruited to DNA double-strand breaks? If so, what is the upstream genetic pathway?

Thanks for the comment. As shown in Figures below, RNF19A was located both in nucleus and cytoplasm with no co-localization with γ -H2AX and DNA damage didn't alter RNF19A nuclear localization significantly.

Figure legend: U2OS cells were treated with IR (2 Gy, 1 h) and subjected to immunofluorescence with RNF19A and γ -H2AX antibody.

2) The authors presented here in the manuscript that RNF19A has two RING domains. However, it is unclear which RING domain confers the catalytic activity. Is there a rationale behind mutating the specific RING domain in this study? There is also a discrepancy in the domain nomenclature compare to NCBI. I assume that the RING2 is referring to the IBR, half RING-finger domain. Comparing the sequences between

the two RING domains may provide more information on their structure. Also, *in vitro* ubiquitination experiments should be included to support the *in vivo* finding.

Thanks for the comment. The mutation of the specific RING domain (Fig.2d) is according to previously reported article⁷. Similar to canonical RINGs, the RING1 domain of the RBR binds an E2~Ub. However, RING2 are dissimilar to RINGs and are instead structurally similar to the IBR domain. Most importantly, all RING2 sequences contain a conserved cysteine residue, the third cysteine in RING2, serving as the active site to which Ub is attached^{8,9}. That's why they have a more appropriate naming-a Ract (required -for-catalysis) domain. We found that deletion of either RING1 or RING2 abolished the RNF19A's regulation of HR (Fig.2i and Supplementary Fig.3b), while deletion of IBR had partial effect (Supplementary Fig.3j).

2d

The ubiquitination of BARD1 by RNF19A was confirmed by *in vitro* ubiquitination assays using purified GST-RNF19A and His-BARD1(aa26-327) proteins. As shown in Figure 4C, WT GST-RNF19A promoted His-BARD1 ubiquitination in the presence of recombinant E1, E2 (UbcH7), ubiquitin proteins *in vitro*. The C316A mutation markedly reduced His-BARD1 ubiquitination *in vitro*.

4c

3) The protein-protein interaction experiments were performed in cells. It is unclear if they interact directly. *In vitro* pull-down assay should be included to show whether the RING domain alone is sufficient for their interaction.

Thanks for the suggestion. In the revised manuscript, we performed *in vitro* pull-down assays using GST-RNF19A and His-BARD1 (aa26-327) purified from bacteria.

The GST pull-down assays showed that GST-RNF19A interacted with His-BARD1 *in vitro*, which was disrupted by deletion of the RING1 domain of RNF19A (R1), indicating that RNF19A directly interacts with BARD1 through RING1 (Fig.2h).

2h

4) The authors showed that GFP-RNF19A overexpression impairs the BRCA1-BARD1 binding. Does RNF19A depletion enhance BARD1-BRCA1 interaction?

As shown in Fig.4g, depletion of RNF19A could enhance BARD1-BRCA1 interaction (lane 3).

4g

5) Since RNF19A is important in regulating the BRCA1-BARD1. It would be interesting to see the RNF19A expression profile throughout the cell cycle and whether the RNF19A-mediated BRCA1-BARD1 function is cell cycle-dependent?

We examined the protein level of RNF19A in different stages of the cell cycle. As shown in Supplementary Fig.4g, RNF19A exhibited consistent expression levels across the cell cycle. To answer whether RNF19A-regulated BARD1 ubiquitination is cell cycle dependent, we examined the ubiquitination levels of BARD1 at different stages of cell cycle. As shown in Supplementary Fig.4f, BARD1 ubiquitination levels were comparable at different cell cycle phases in control and RNF19A-depleted cells.

S4g

S4f

Minor comments:

1) Some of the labels are inconsistent between text and figures, such as XRCC vs XRCC4; RPA vs RPA32? It could be confusing to the general reader. Also for ubiquitination vs ubiquitylation, even though they are interchangeable.

Thanks for the suggestion. We've unified these words to XRCC4, RPA32 and ubiquitination.

2) A precise detailed information of the deletion used in the study should be provided.

In the revised manuscript, we added the detailed sequence of the deletion used in this study, including RNF19A-R1 (Δ 132-179), R2 (Δ 199-264), R3 (Δ 301-332) and BARD1-B1 (Δ 1-138), B2 (Δ 424-777), B3 (Δ 568-777), NES (Δ 92-120).

3) Figure 6A labels are off.

Thanks for pointing out this mistake. We've corrected the labels.

4) It would be helpful to provide a little bit more information for the RNF19A background in the revised manuscript.

Thanks for the suggestion. In the revised manuscript, we added a brief background of RNF19A in the last paragraph of the introduction section.

Reviewer #3 (Remarks to the Author):

In this report, the authors proposed a model in which RNF19A may play a role to suppress HR activity for the fine-tuning of DSB repair. The authors showed that depletion of RNF19A enhances the proportion of DSBs undergoing end resection, which directs the repair pathway toward HR. Furthermore, the authors demonstrated that BARD1 is ubiquitylated by RNF19A, and that this ubiquitylation step is associated with BRCA1 stability and BARD1 nuclear export; hence, the authors proposed a model in which RNF19A suppresses HR at the step of resection by ubiquitylating BARD1. Some of their findings are interesting; however, several data are technically preliminary. More importantly, I am not convinced by the interpretation proposed by these authors, as follows. 1) The authors stated that RNF19A suppresses HR and that its depletion enhances HR; why does RNF19A need to suppress the accurate HR, i.e., what is the advantage of HR suppression for cells? If the restored HR pathway is still accurate, under RNF19A deficiency, cells do not require RNF19A.

HR is presumed to be predominantly a non-mutagenic and precise type of repair. The over-suppression of BRCA1/BARD1 function by RNF19A amplification might cause genomic instability and promote tumorigenesis. However, hyperactivation of the BRCA-HR pathway could also cause genomic instability. As the reviewer pointed out in point 2 below, hyperactivation of this pathway could increase SSA, which is more mutagenic. The BRCA1 associated factor RAP80 helps to restrict excessive end resection and HR function of BRCA1, as excessive end resection could cause genomic instability¹⁰. It has also been shown that BRCA1 PARylation can fine-tune its activity, and failed BRCA1 PARylation led to failed BRCA1 release from chromatin and from IRIF, resulting in the development of high and deregulated HRR activity, marked genome instability, and evidence of aneuploidy¹¹. Also BRCA1 overexpression is associated with aneuploidy and poor prognosis in breast cancers¹². In a similar vein, the negative regulation of BARD1 by RNF19A might be important for maintaining genomic stability. Thus, under normal conditions, RNF19A acting as a fine-tuning coordinator in BRCA1/BARD1 complex. It is also conceivable that under some circumstance the BRCA1/BARD1 pathway is suppressed such as at high-repeat regions, and cells could use the negative regulators such as RNF19A. Also, under pathological conditions, RNF19A could be exploited by oncogenic processes to suppress HR and promote cell transformation. This requires further investigated to understand how RNF19A is regulated.

2) The authors reported that the enhanced resection observed in RNF19A-depleted cells is dependent on BARD1; however, what is the mechanism underlying the BARD1-dependent resection? Although the authors referred to references 33 and 34, I do not think that these studies demonstrate the mechanism underlying the manner in which BARD1 promotes resection. Is it via the activation of the initiation of resection by CtIP/MRN or via the removal of the 53BP1 barrier by BRCA1? Overall, this conclusion remains preliminary and further investigation is required to elucidate this issue.

As an obligatory partner of BRCA1, BARD1 promotes retention of the BRCA1/BARD1 complex at DNA damage sites¹³⁻¹⁷, which is essential for the process of HR, including RAD51 recruitment and DNA end resection¹⁸. Many studies have shown that BRCA1-BARD1 is an accessory factor of DNA end resection, but is not an indispensable resection machinery^{19,20}. The BRCA1-BARD1 has a role in the nucleolytic resection of DSB ends, as first proposed on the basis of the cell cycle-dependent association of BRCA1 with the end resection factor CtBP-interacting protein (CtIP)²¹⁻²³. BRCA1 promotes end resection via its interaction with CtIP/MRN and counteracting the inhibitory effect of 53BP1. Suppression of either BRCA1 or BARD1 led to a great reduction of end resection and HR efficiency. Other studies suggest that while 53BP1 could partially overcome the HR defect associated with BRCA1 deficiency, it did not suppress the HR defect in BARD1-deficient cells¹⁸. Also, depletion of BARD1 didn't affect the 53BP1 focus formation¹⁴. In the revised manuscript, we examined CtIP protein level and focus formation in BARD1 knockdown cells, and found that CtIP foci were extensively suppressed in BARD1-depleted cells compared to the control group (Results are shown in Point 4 below). On the other hand, depletion of 53BP1, an inhibitor of DNA end resection, didn't suppress the end resection defect in BARD1-deficient cells (Results are shown in Point 5 below). These results indicate that in RNF19A depleted cells, BARD1/BRCA1 can promote end-resection/HR at least partially through CtIP.

1. Major technical issue:

the data presented in Figure 1 are confusing. The authors showed a faster DSB repair in cells treated with the RNF19A siRNA at 8 h after IR. I assume that the authors aimed to state that the enhanced HR activity accelerates DSB repair. However, this is unlikely to occur because HR is a slower repair pathway compared with NHEJ, i.e., the switch of repair pathway from NHEJ to HR delays the speed of the DSB repair kinetics. Hence, generally, the increase in the fraction of DSBs undergoing HR yields greater γ H2AX foci compared with control cells. In the reporter assays depicted in Figure 1F-G, strangely, RNF19A depletion did not change NHEJ, even though HR was enhanced. When the HR pathway is enhanced, some other pathways should be downregulated. Which repair fraction is converted into HR? In normal human cells, the majority of DSBs are repaired by either NHEJ or HR. SSA has a minor impact on the DSB fraction in normal cells. Thus, I am not convinced by these data.

We agree with the reviewer that HR is a slower repair pathway. However, HR might be important for complex breaks not easily fixed by NHEJ. CtIP has been reported to be especially important for processing complex DNA ends and perform end resection²⁴. Therefore, the γ H2AX foci at later time points might reflect those complex DNA damages and need BRCA/MRE11/CtIP end resection to repair. This is why we observed less γ H2AX foci at later time points in RNF19A depleted cells and more sustained γ H2AX foci in RNF19A overexpressing cells. Other studies also observed similar pattern of γ H2AX when HR pathway is compromised^{2,25-27}.

We think RNF19A is a regulator to fine tune HR, not an essential factor for HR. When the effect on NHEJ was evaluated, especially using the reporter, such an effect

is not evident, especially for simple breaks that can easily be joined by NHEJ. For examples, ZMYM3²⁸, which was reported to regulate BRCA1-A complex; Mettl3²⁵, which could catalyze m6A modification of RNAs at DSBs, and USP15²⁶, a deubiquitinating enzyme for BARD1, also only affects HR but not NHEJ.

To better study how RNF19A affects HR/NHEJ balance, we used CLZ3 cell line, containing a dual-reporter, for the simultaneous measurement of both HR and NHEJ efficiency at the same chromosomal site⁴. In this system, we detected an increased HR (tdTomato+) efficiency in RNF19A knockdown cells, while the NHEJ (GFP+) efficiency was mildly compromised.

Last, why did the authors use different time points and IR doses to examine DDR foci in Figure 1i? In the representative image, the authors showed >20-30 RPA/RAD51 foci (as far as I can see) at 3-5 h after 2 Gy. The number of RAD51 foci in cells treated with the RNF19A siRNA is clearly greater than that in control cells in this figure. However, in Figure 1A, the percentage of RNF19A-siRNA-treated cells with >10 γ H2AX foci per cell at 8 h is ~10%-20%. I do not understand how the RAD51 foci are quickly repaired within 3 h (5-8 h post IR). The authors should perform the time-course experiment and show the number of foci, rather than the percentage of foci, as shown in other figures.

Thanks for the suggestion. We used 2 Gy to detect RAD51 and RPA foci while 1Gy was used for examining γ -H2AX, so there may be discrepancy when comparing the results in Fig. 1a and i. In the revised manuscript, we uniformed the IR dosage (1Gy) and recalculated the raw numbers of foci per cell for foci quantification, as shown in Fig. 1b and i, RNF19A depletion resulted in a significantly decreased γ H2AX foci at late time points (8 and 24 h) and elevated accumulation of BRCA1, BARD1, RAD51 and RPA32 focus formation.

As for the inconsistency in time points, in our hand, 1-2 h didn't give strong signals for RPA and RAD51, therefore, we used a relative longer time point to observe a more visible RPA and RAD51 foci, which was also used by other researchers^{1,2,26}.

In addition, we performed the time-course experiment to examine RAD51 foci and quantify the number of foci per cell as shown in Figures below, the increase of RAD51 foci was more pronounced in RNF19A-knockdown U2OS cells.

Figure legend: Control and RNF19A knockdown U2OS cells were treated with or without IR (2 Gy), RAD51 foci before or 1 h ,8 h and 24 h after IR were detected by immunofluorescence. Nuclei were visualized with DAPI (blue). Representative images are shown in (a). Quantification of focus signals per cell ($n \geq 100$) is shown in (b). Scale bars, 10 μ M.

2. Depletion of HR blockers, such as RAP80, increases resection, which leads to an abnormal repair pathway (PMID: 21406551). I am surprised that the increase in unscheduled resection in RNF19A-depleted cells still conferred normal HR. To clarify the quality of HR and resection-mediated repair in RNF19A-depleted cells, the authors should examine SCEs and SSA.

Thanks for the constructive suggestion. Since end resection is a common step for HDR and single strand annealing (SSA), a defect will compromise both pathways²⁹. In the revised manuscript, we used SA-GFP to detect single-strand annealing (SSA) efficiency in control and RNF19A knockdown cells. As shown in Supplementary Fig.3o, RNF19A-depleted cells also led to a mildly increased efficiency of SSA, suggesting RNF19A has a role in homologous repair before the branch point of HDR and SSA.

We also examined the frequency of sister chromatid exchange (SCE) in WT and RNF19A KO U2OS cells treating with IR. As shown in Supplementary Fig.3p and q,

depletion of RNF19A led to an increased frequency of SCE, which indicates the enhancing HR activity.

3. The authors analyzed the cell-cycle distribution of RNF19A-depleted cells, stating that the depletion did not affect the cell-cycle status. However, in Figure 3A, B, the authors showed that almost all cells harbored RPA/RAD51 foci in RNF19A depleted cells. Does this mean that the RNF19A siRNA causes HR in G1 cells as well? The authors should perform an analysis of the foci using appropriate cell-cycle markers, such as CENPF/Cyclin A and BrdU/EdU, and should report the number of foci at each phase of the cell cycle.

Thanks for the suggestion. In the revised manuscript, we examined RPA/RAD51 foci co-immunostained with cyclin A-an established marker of the S and G2 phases. As shown in Fig.3a-d, RPA and RAD51 foci only formed in cells with cyclin A staining, indicating RNF19A only affect RPA/RAD51 foci in S/G2 phase.

4. Depletion of BARD1 causes BRCA1 degradation. In general, it is known that the presence of BRCA1 contributes to the stabilization of BRCA1-complex proteins, including CtIP. Thus, the authors should determine the level of the CtIP protein in BARD1-depleted cells.

Thanks for the suggestion. In the revised manuscript, we examined CtIP protein level in U2OS cells transfected with control or BARD1 siRNA. As shown in Supplementary Fig. 3n, CtIP protein level didn't change much in BARD1 depleted cells compared to control cells. We also examined CtIP foci in these cells and found a decreased level of CtIP foci in BARD1 knockdown cells (Supplementary Fig. 3l and m).

5. In Figure 3, the authors stated that the enhanced resection detected in RNF19A-depleted cells is dependent on BARD1. To clarify whether it is dependent on the BRCA1/53BP1 axis (PMID: 27239795), the authors should investigate whether 53BP1 depletion restores resection in the BARD1-depleted background.

Thanks for the constructive suggestion. In the revised manuscript, we assessed the resection efficiency in BARD1 knockdown or BARD1 and 53BP1 double-knockdown ER-AsiSI U2OS cells. As shown in Supplementary Fig.3k, depletion of 53BP1 didn't rescue the end resection defect in BARD1-deficient cells.

6. In Figure 4, the authors showed that RNF19A ubiquitylates BARD1. Is this ubiquitylation induced after DNA damage? Or is it independent of DNA damage?

In the revised manuscript, we investigated whether RNF19A regulates BARD1 ubiquitination after DNA damage. As shown in Supplementary Fig. 4a, RNF19A's ubiquitination of BARD1 decreased after DNA damage.

S4a

7. In lines 250-251, the authors stated that “RNF19A is involved in nuclear-to-cytoplasmic shuttling of BARD1 in response to DNA damage”. The enrichment of nuclear BARD1 in immunoblotting was marginal after IR (lanes 1 and 2 in Figure S5C). Is this enrichment cancelled by the RNF19A siRNA?

In the revised manuscript, we quantified the nuclear and cytoplasmic BARD1 protein level in control and RNF19A knockdown cells. As shown in Supplementary Fig.5h, IR treatment induced more nuclear BARD1, and this effect was cancelled by RNF19A depletion, even though RNF19A-depleted cells have more BARD1 in the nucleus under both IR-treated or untreated conditions.

The authors should also show the enrichment of nuclear BARD1 in response to DNA damage by immunofluorescence, i.e. before and after IR, which should be feasible because the authors used immunofluorescence in other experiments.

Thanks for the suggestion. In the revised manuscript, we added the results of the enrichment of nuclear BARD1 and BRCA1 by IF under both IR-treated or untreated conditions. As shown in Supplementary Fig.5a, a stronger cytoplasmic fraction of BARD1, but not BRCA1 (although nuclear BRCA1 was decreased, resulting in a small

decreased in nuc/cyto ratio) observed upon RNF19A over-expression.

8. In lines 251-253, the authors mentioned that “Previous studies have demonstrated that BRCA1 protein levels increase in late G1 and reach a maximum in S phase, while BARD1 is expressed at parallel levels throughout the cell cycle.” However, I could not detect an obvious alteration in the level of the BRCA1 protein level between the G1 and S/G2 phases under DTB-dependent cell-cycle synchronization. To demonstrate that cells are successfully synchronized, the percentage of the cell-cycle phase should be shown at each time point.

We repeated the experiment by using anti-BRCA1 antibody for endogenous IP. BRCA1 expression was increased in S phase. Overexpression of RNF19A decreased the interaction between BRCA1 and BARD1 in S phase. Cell cycle synchronization was through releasing cells from double thymidine (Supplementary Fig.4e).

9. In the discussion, the authors stated that “The fine-tuning of BRCA1/BARD1 function by RNF19A-mediated ubiquitination might cause genomic instability and promote tumorigenesis.” I wonder whether this should be “The lack of fine tuning of BRCA1/BARD1 function...”. The authors should clarify whether they discuss the situation in control cells, in RNF19A-overexpressing cells, or in tumor cells.

Thanks for the constructive suggestion. We’ve modified the wording in the discussion.

References

- 1 Gao, M. *et al.* USP52 regulates DNA end resection and chemosensitivity through removing inhibitory ubiquitination from CtIP. *Nature communications* **11**, 5362, doi:10.1038/s41467-020-19202-0 (2020).
- 2 Zhou, Q. *et al.* The bromodomain containing protein BRD-9 orchestrates RAD51-RAD54 complex formation and regulates homologous recombination-mediated repair. *Nature communications* **11**, 2639, doi:10.1038/s41467-020-16443-x (2020).
- 3 Zhao, F. *et al.* ASTE1 promotes shieldin-complex-mediated DNA repair by attenuating end resection. *Nature cell biology*, doi:10.1038/s41556-021-00723-9 (2021).
- 4 Chen, Y. *et al.* A PARP1-BRG1-SIRT1 axis promotes HR repair by reducing nucleosome density at DNA damage sites. *Nucleic acids research* **47**, 8563-8580, doi:10.1093/nar/gkz592 (2019).
- 5 Brzovic, P. S., Lissounov, A., Christensen, D. E., Hoyt, D. W. & Klevit, R. E. A UbcH5/ubiquitin noncovalent complex is required for processive BRCA1-directed ubiquitination. *Molecular cell* **21**, 873-880, doi:10.1016/j.molcel.2006.02.008 (2006).
- 6 Daza-Martin, M. *et al.* Isomerization of BRCA1-BARD1 promotes replication fork protection. *Nature* **571**, 521-527, doi:10.1038/s41586-019-1363-4 (2019).
- 7 Rivkin, E., Kierszenbaum, A. L., Gil, M. & Tres, L. L. Rnf19a, a ubiquitin protein ligase, and Psmc3, a component of the 26S proteasome, tether to the acrosome membranes and the head-tail coupling apparatus during rat spermatid development. *Developmental dynamics : an official publication of the American Association of Anatomists* **238**, 1851-1861, doi:10.1002/dvdy.22004 (2009).
- 8 Wenzel, D. M., Lissounov, A., Brzovic, P. S. & Klevit, R. E. UBCH7 reactivity profile reveals parkin and HHARI to be RING/HECT hybrids. *Nature* **474**, 105-108, doi:10.1038/nature09966 (2011).
- 9 Dove, K. K., Stieglitz, B., Duncan, E. D., Rittinger, K. & Klevit, R. E. Molecular insights into RBR E3 ligase ubiquitin transfer mechanisms. *EMBO reports* **17**, 1221-1235, doi:10.15252/embr.201642641 (2016).
- 10 Hu, Y. *et al.* RAP80-directed tuning of BRCA1 homologous recombination function at ionizing radiation-induced nuclear foci. *Genes & development* **25**, 685-700, doi:10.1101/gad.2011011 (2011).
- 11 Hu, Y. *et al.* PARP1-driven poly-ADP-ribosylation regulates BRCA1 function in homologous recombination-mediated DNA repair. *Cancer discovery* **4**, 1430-1447, doi:10.1158/2159-8290.CD-13-0891 (2014).
- 12 Vohhodina, J. *et al.* RAP80 and BRCA1 PARsylation protect chromosome integrity by

- preventing retention of BRCA1-B/C complexes in DNA repair foci. *Proceedings of the National Academy of Sciences of the United States of America* **117**, 2084-2091, doi:10.1073/pnas.1908003117 (2020).
- 13 Becker, J. R. *et al.* BARD1 links histone H2A Lysine-15 ubiquitination to initiation of BRCA1-dependent homologous recombination. *bioRxiv*, 2020.2006.2001.127951, doi:10.1101/2020.06.01.127951 (2020).
- 14 Dai, L. *et al.* Structural insight into BRCA1-BARD1 complex recruitment to damaged chromatin. *Molecular cell*, doi:10.1016/j.molcel.2021.05.010 (2021).
- 15 Wu, W. *et al.* Interaction of BARD1 and HP1 Is Required for BRCA1 Retention at Sites of DNA Damage. *Cancer research* **75**, 1311-1321, doi:10.1158/0008-5472.CAN-14-2796 (2015).
- 16 Becker, J. R. *et al.* BARD1 reads H2A lysine 15 ubiquitination to direct homologous recombination. *Nature*, doi:10.1038/s41586-021-03776-w (2021).
- 17 Deng, M., Hou, J. & Lou, Z. Bivalent recognition of histone marks by BARD1. *Trends in cell biology*, doi:10.1016/j.tcb.2021.06.005 (2021).
- 18 Zhao, W. *et al.* BRCA1-BARD1 promotes RAD51-mediated homologous DNA pairing. *Nature* **550**, 360-365, doi:10.1038/nature24060 (2017).
- 19 Cruz-Garcia, A., Lopez-Saavedra, A. & Huertas, P. BRCA1 accelerates CtIP-mediated DNA-end resection. *Cell reports* **9**, 451-459, doi:10.1016/j.celrep.2014.08.076 (2014).
- 20 Alagoz, M. *et al.* SETDB1, HP1 and SUV39 promote repositioning of 53BP1 to extend resection during homologous recombination in G2 cells. *Nucleic acids research* **43**, 7931-7944, doi:10.1093/nar/gkv722 (2015).
- 21 Densham, R. M. *et al.* Human BRCA1-BARD1 ubiquitin ligase activity counteracts chromatin barriers to DNA resection. *Nature structural & molecular biology* **23**, 647-655, doi:10.1038/nsmb.3236 (2016).
- 22 Tarsounas, M. & Sung, P. The antitumorigenic roles of BRCA1-BARD1 in DNA repair and replication. *Nature reviews. Molecular cell biology* **21**, 284-299, doi:10.1038/s41580-020-0218-z (2020).
- 23 Batenburg, N. L. *et al.* CSB interacts with BRCA1 in late S/G2 to promote MRN- and CtIP-mediated DNA end resection. *Nucleic acids research* **47**, 10678-10692, doi:10.1093/nar/gkz784 (2019).
- 24 Wilkinson, O. J. *et al.* CtIP forms a tetrameric dumbbell-shaped particle which bridges complex DNA end structures for double-strand break repair. *eLife* **8**, doi:10.7554/eLife.42129 (2019).
- 25 Zhang, C. *et al.* METTL3 and N6-Methyladenosine Promote Homologous Recombination-Mediated Repair of DSBs by Modulating DNA-RNA Hybrid Accumulation. *Molecular cell* **79**, 425-442 e427, doi:10.1016/j.molcel.2020.06.017 (2020).
- 26 Peng, Y. *et al.* The deubiquitylating enzyme USP15 regulates homologous recombination repair and cancer cell response to PARP inhibitors. *Nature communications* **10**, 1224, doi:10.1038/s41467-019-09232-8 (2019).
- 27 Peng, M., Litman, R., Jin, Z., Fong, G. & Cantor, S. B. BACH1 is a DNA repair protein supporting BRCA1 damage response. *Oncogene* **25**, 2245-2253, doi:10.1038/sj.onc.1209257 (2006).
- 28 Leung, J. W. *et al.* ZMYM3 regulates BRCA1 localization at damaged chromatin to promote DNA repair. *Genes & development* **31**, 260-274, doi:10.1101/gad.292516.116 (2017).
- 29 Chen, C. C. *et al.* ATM loss leads to synthetic lethality in BRCA1 BRCT mutant mice associated

with exacerbated defects in homology-directed repair. *Proceedings of the National Academy of Sciences of the United States of America* **114**, 7665-7670, doi:10.1073/pnas.1706392114 (2017).

REVIEWERS' COMMENTS

Reviewer #1 (Remarks to the Author):

The additional experiments carried out by the authors satisfactorily address all the major questions/concerns. It is done very well.

Reviewer #2 (Remarks to the Author):

The authors have satisfactorily addressed the concerns I raised. I support the publication of this study.

Reviewer #3 (Remarks to the Author):

The authors adequately answered my concerns. The manuscript has been well improved. However, it is very strange that the images of BRCA1 and BARD1 in knockdown cells are identical (Figure 1h) because the authors used the other area for BRCA1/BARD in control cells. Images of other fields should be shown to avoid any error or suspicion of fabrication.

REVIEWERS' COMMENTS

Reviewer #1 (Remarks to the Author):

The additional experiments carried out by the authors satisfactorily address all the major questions/concerns. It is done very well.

We thank the reviewer for the positive comments.

Reviewer #2 (Remarks to the Author):

The authors have satisfactorily addressed the concerns I raised. I support the publication of this study.

We thank the reviewer for the positive comments.

Reviewer #3 (Remarks to the Author):

The authors adequately answered my concerns. The manuscript has been well improved. However, it is very strange that the images of BRCA1 and BARD1 in knockdown cells are identical (Figure 1h) because the authors used the other area for BRCA1/BARD in control cells. Images of other fields should be shown to avoid any error or suspicion of fabrication.

We thank the reviewer for the positive comments. We usually coimmunostained BRCA1 and BARD1 as most foci of BRCA1 and BARD1 were co-localized, which makes them look like identical. We have used co-staining of BRCA1 and BARD1 throughout the manuscript. Very appreciated for pointing out the inconsistent showing areas between control and knockdown groups. We have chosen the same area of BRCA1 and BARD1 in control cells in the revised manuscript.